# Detection of aerosol and cloud features for the EarthCARE lidar ATLID: the A-FM product

Gerd-Jan van Zadelhoff[1], David P. Donovan[1], and Ping Wang[1]

[1]Royal Netherlands Meteorological Institute (KNMI), de Bilt, the Netherlands

**Correspondence:** Gerd-Jan van Zadelhoff (gerd-jan.van.zadelhoff@knmi.nl)

**Abstract.** The EarthCARE satellite mission's objective is to retrieve profiles of aerosol and cloud physical and optical properties using the combination of cloud-profiling radar (CPR), high spectral resolution UV lidar (ATLID), and passive multi-spectral imager (MSI) data. Based on synergistic retrievals using data from these instruments, the 3D atmospheric cloud/aerosol state is estimated and then used to model the Top-of-Atmosphere (TOA) broad-band radiances, which may then be compared to co-incident EarthCARE broad band radiometer (BBR) measurements. A high spectral resolution lidar enables the independent retrieval of extinction and backscatter but, being space-based, suffers from relatively low signal-to-noise ratio (SNR) levels. The ATLID FeatureMask (A-FM) product provides a feature detection mask for the existence of atmospheric features within the lidar profiles based on a number of (statistical) image reconstruction techniques. Next to this, it also identifies those regions where the lidar beam has been fully attenuated and where the surface backscatter has impacted the measured lidar backscatter signals directly above the surface. From the pixels assigned as clear-sky (with no features present above), the "clear-sky-averaged" profiles for the three ATLID channels, the co-polar Mie channel, the total cross channel and the co-polar Rayleigh channel, are created. These 'feature-free' or 'clear-sky' profiles are useful for e.g. assessing the quality of the ATLID Level 1 (L1) attenuated backscatters. An important goal of the A-FM product is to guide smoothing strategies within downstream processors e.g. the ATLID profile retrieval (A-PRO) algorithm which directly follows A-FM within the EarthCARE Level 2 (L2) processing chain. Within the A-PRO algorithm, profiles of extinction, backscatter and linear depolarization ratio are retrieved. However, smoothing of the ATLID L1 attenuated backscatter is necessary since the SNR levels present at the ATLID native resolution is generally not sufficient for meaningful retrievals to be conducted. At the same time, to prevent biased retrievals, any smoothing procedure must respect the cloud/aerosol structure and avoid mixing strong features, e.g. clouds, and weak features, e.g. aerosol regions, together. The A-FM product provides the A-PRO algorithm with important information that is used to guide various smoothing procedures. To enable the processing of the large data-sets from observation up to L2 retrievals each EarthCARE orbit is separated in to eight frames, divided at latitudes of $22.5^o$N/S and $62.5^o$N/S. As a secondary product, A-FM outputs can be used to conduct a frame-by-frame evaluation of the ATLID L1 cross-talk calibration, where an EarthCARE frame is 1/8 of a full orbit. This evaluation can be performed by comparing the retrieved clear sky profiles to the expected channel profiles. The A-FM product has been applied to both synthetic data from the EarthCARE end-to-end simulator (ECSIM) as well as the L1 data from the Aeolus wind-lidar mission. Comparisons against the ECSIM model truth indicate A-FM has a percentage correctness $>90\%$ and is capable of reliably detecting aerosol and cloud regions with extinctions $> 10^{-5}$ m$^{-1}$.

# 1 Introduction

The EarthCARE mission (Earth Clouds, Aerosols and Radiation Explorer; (Illingworth et al., 2015) is a collaborative Earth observation satellite. The responsible bodies are the European Space Agency (ESA) and the Japan Aerospace Exploration Agency (JAXA). The satellite is planned to be launched in 2024 and its primary mission is to improve the understanding of the interaction between clouds, aerosols and atmospheric radiation, and how these interactions affect climate and weather. EarthCARE will fly in a sun-synchronous orbit, with a descending node crossing time of 14:00 hours, an inclination of $97^o$, revisit time of 25 days and at an altitude of 393 km. The platform is comprised of a 94 GHz Doppler cloud profiling radar (CPR), a 355 nm high spectral resolution atmospheric lidar (ATLID), a multispectral imager (MSI), and a broadband long- and short-wave radiometer (BBR). EarthCARE science is built around the synergistic use of these four advanced sensors (Eisinger et al., 2022), where ATLID, CPR and MSI data are combined in order to estimate the 3D atmospheric properties of clouds, aerosols and precipitation, including their optical and microphysical properties. Forward modelled radiative properties of the retrieved 3D atmospheric fields can subsequently be compared to the BBR measurements for near-real time evaluation of the performed retrievals (Barker et al., 2022). In order to achieve these aims, a chain of individual instrument geophysical algorithms (L2a) and synergistic (e.g. multi-instrument algorithms; dubbed as L2b algorithms) have been developed (Eisinger et al., 2022). All the EarthCARE algorithms are realized as standalone processors, but are designed to fit into the overall retrieval process as their outputs are used as high level inputs, i.e. a priori settings, for algorithms present later in the chain.

A lidar feature detection mask is a mask used to identify different atmospheric features, such as clouds or aerosols. Next to this it also identifies regions where the lidar beam has been fully attenuated or when the measured backscatter in a range bin is affected by the ground surface backscatter. For lidar instruments which suffer from relatively low signal-to-noise ratios (SNR), which includes all space missions, some type of "feature mask" is commonly developed and applied. These masks are needed to provide context to the lidar signals. In this paper, the ATLID FeatureMask(A-FM) L2a retrieval algorithm developed for the ATLID instrument is described, as well as the evaluation using synthetic model fields and data from the space-based Doppler wind lidar from the ESA Earth Explorer Aeolus Mission (Reitebuch et al., 2019).

The Cloud-Aerosol Lidar and Infrared Pathfinder Satellite Observations (CALIPSO; Winker et al., 2009) was launched on April 28, 2006 in order to study the impact of clouds and aerosols on the Earth's radiation budget and climate. The CALIPSO lidar, the Cloud-Aerosol LIdar with Orthogonal Polarization (CALIOP), is an elastic backscatter lidar that emits linearly polarized laser light at 532 and 1064 $nm$ and receives both the linear polarized signals and the cross polarized signals at 532 $nm$.

For the CALIPSO lidar (Winker et al., 2009), a number of similar feature-mask type algorithms have been created. Two of the products created are described here as reference. The first is the Vertical Feature Mask (VFM; Vaughan et al., 2009; Liu et al., 2019; Avery et al., 2020; Kim et al., 2018), currently at Version 4-21, which describes the vertical and horizontal distribution of cloud and aerosol layers observed by the CALIOP lidar. The VFM mask discriminates aerosols and clouds based on their physical feature differences within an (averaged) profile. The need to discriminate aerosols and clouds requires relatively large horizontal averaging windows of profiles at native resolution $\frac{1}{3}$ $km$ up to 80 $km$. The second CALIOP algorithm is more recent, and retrieves "context-sensitive" features within the lidar signals using 2-D image information from neighboring lidar

profiles (Vaillant de Guélis et al., 2021). It uses the backscatter signals from all three available CALIOP channels and iteratively determines lower thresholds to find weaker features within the image. The main advantage of this method is that the complex shapes of aerosol and cloud features are better preserved and masked. Even though the method implemented is different the basic idea of using image reconstruction techniques is similar to the method described in this paper.

Another new approach has been created for the NASA ICESat-2 mission, which carries the Advanced Topographic Laser Altimeter System (ATLAS; Markus et al., 2017) operating at 532 nm. The aim of this mask (Herzfeld et al., 2021) is to detect layers in the ICESat-2 data during complex atmospheric situations, specifically aiming at the detection of blowing snow and thin cirrus clouds. The method adopts a Gaussian radial data aggregation function with an auto-adaptive threshold determination.

The A-FM provides a probability mask of whether a pixel contains cloud and/or aerosols, it does not perform any typing information of the respective pixel. The main goal for both the 2-D approaches for CALIOP and ATLAS are the same as that for the FeatureMask detection algorithm described within this paper. The VFM product provides typing information of the aerosol and cloud returns, which requires a far higher signal to noise ratio (SNR). The VFM product combines the results from the feature finder algorithm SIBYL (Vaughan et al., 2009), whereas the typing is determined by three independent scene classification algorithms; the cloud and aerosol discrimination algorithm, the aerosol sub-typing algorithm and the cloud phase discrimination algorithm (Liu et al., 2019; Avery et al., 2020; Kim et al., 2018). Both CALIPSO and ICESAT-2 operate elastic backscatter lidars, whereas ATLID is a high spectral resolution lidar (HSRL). To benefit from the ATLID HSRL ability to directly retrieve extinction and backscatter separately, correct averaging of the data is essential. The cloud and aerosol classification of any identified features are performed at a later stage in the EarthCARE L2 chain.

To help guide the smoothing procedures performed in the ATLID profile retrieval algorithm (A-PRO; Donovan et al., 2022) a mask at the highest possible resolution is needed where strong (liquid layers, optically thick ice clouds, optically thick aerosol regimes and surface returns) and weak back-scattering regions (aerosol fields and thin ice clouds) are distinguished from each other and from clear sky regions. This ensures that backscatter signals from liquid clouds are not mixed with aerosol or cirrus layers.

Within the A-PRO algorithm, profiles of extinction, backscatter and depolarization are retrieved for which the ATLID attenuated backscatter signal-to-noise ratios are insufficient at the native ATLID resolution. Smoothing of the ATLID signals is necessary in order to increase their associated SNRs. Indiscriminately smoothing signals, within A-PRO, however, will result in incorrect retrievals which do not represent the actual atmospheric state leading to, e.g. an incorrect target classification. The A-FM product provides the A-PRO algorithm with a field of significant detection. This field is used within A-PRO to guide local smoothing strategies that aim to ensure that strong and weak attenuated backscatter signals are not mixed together and not diluted by smoothing clear sky values.

The A-FM processor is the first processor in the level-2 processing chain to be applied to the ATLID data and thus its output is important for the whole ATLID processing chain. In particular, A-FM output is used by A-PRO and the output is A-PRO is used to generate the synergistic lidar-radar target classification (Irbah et al., 2023) product which, in turn, is ingested by the synergistic cloud and aerosol property retrieval algorithm (ACM-CAP) (Mason et al., 2022).

For the testing of all the EarthCARE processors, a number of detailed simulated scenes have been created, using as input, a number of atmospheric states calculated by numerical weather prediction (NWP) models. These model states were subsequently transformed into EarthCARE Simulator scenes after which realistic attenuated backscatter (L1b) signals and associated errors for the three ATLID channels were calculated. For a full description of these scenes see Donovan et al. (2023). Within this paper, we focus on the so–called 'Halifax-scene' (Figure 8) and a sub-set of this scene focusing on the aerosol regime (the Halifax-aerosol scene; Figure 1) for the evaluation of the A-FM processor.

In Section 2, we provide a detailed description of the FeatureMask detection of areas with aerosol and/or cloud particles. In Section 3 the performance and sensitivity of the procedure is described using simulated EarthCARE L1b data and corresponding model truth from the test scenes.

In August 2018 the European Space Agency (ESA) launched the Aeolus Earth Explorer Mission (Reitebuch et al., 2019), carrying the first space-based Doppler wind lidar. The main instrument on board is an ultraviolet (UV) high-spectral resolution lidar, the Atmospheric LAser Doppler INstrument (ALADIN). Even though the focus of Aeolus is on the retrieval of line–of–sight winds the instrument measures atmospheric profiles of Mie and Rayleigh attenuated backscatter signals. Both A-FM and A-PRO developed for the inversion of ATLID signals have been adapted for the Aeolus mission. These Aeolus specific versions, named AEL-FM and AEL-PRO respectively, are currently part of the Aeolus operational processing stream. In the case of AEL-FM, the main difference with respect to A-FM has to do with oversampling the 24 vertical bins of Aeolus data to a higher vertical resolution in order to enable the use of the A-FM procedures described below. Once the input has been regridded, AEL-FM and A-FM are in essence similar, and only once the output has been created at high resolution it needs to be remapped to the Aeolus measurement grid. In Section 4 the results for two simulated tests scenes and one Aeolus-CALIOP collocated orbit are presented. Finally, the conclusions are presented in Section 5.

## 2 FeatureMask Retrieval algorithm

### 2.1 Algorithm background

The A-FM algorithm is used to determine the feature detection mask based on exploiting the time-height correlation of the attenuated backscatter data while using a minimum number of hard coded or input dependent thresholds. This approach enables the retrieval to deal with the low signal to noise ratios associated with ATLID signals at the instrument resolution (single-pixel level). Since A-FM retrievals are used to guide smoothing techniques and windows in later processors, the first goal is to separate 'strong' features from 'weaker' features. Two complementary methods are employed in the algorithm to retrieve the feature mask: the median-hybrid method (Russ, 2007, Chapter 4) for strong features (Section 2.5), and a data smoothing strategy based on a simplified maximum entropy method (Smith and Grandy, 1985) for the detection of weaker features (Section 2.6). It was found that employing the full maximum entropy method is both too time consuming and does not always converge to a single optimal smoothed image. The maximum entropy method was deemed to focus too much on the stronger features in the noise while missing some of the more tenuous widespread aerosol layers. To ensure that the algorithm is robust and fast enough for the usage of space-based data, the algorithm has been simplified and now uses 4 pre-defined convolved

images, instead of attempting to retrieve the rigorous, maximum entropy defined image. These four convolved images span the entire parameter space in which any optimum maximum entropy image has been identified in the evaluation period of the algorithm. Based on these two methods, coherent atmospheric structures are defined, dubbed as features, within this work.

The lidar deployed on the EarthCARE satellite (ATLID) is a high-spectral-resolution (HSRL) depolarization lidar operating at a wavelength of 355 nm. The instrument emits linearly polarized laser pulses at a rate of 51 Hz with a pulse energy of 31-35 mJ. The output beam has a divergence of 36 $\mu$rad and points $3^o$ backwards in order to minimize specular reflection by ice cloud particles. The laser beam is collected by a 62 cm diameter telescope and separated into three receiver channels. The incoming signals first pass through a polarised beam splitter separating the cross-polar signals from the co-polar signals. The co-polar contribution in the return signal is subsequently separated into contributions from the thermally broadened molecular (Rayleigh) return and the spectrally narrow elastic backscatter returns from cloud and/or aerosol particles by means of a Fabry-Perot Etalon based spectralfilter.

Within the EarthCARE terminology, the former signal is referred to as the co-polar Rayleigh return and the latter as the co-polar Mie return. The signals from each channel are detected by Memory Charge-Coupled Devices (MCCDs) allowing for single photon detection. The vertical resolution is 103 m up to 20 km altitude and about 500 m up to 40 km altitude, with an effective along-track spatial resolution of about 280 m (after onboard integration of two consecutive lidar profiles). The profile signals will experience a vertical crosstalk of 11% up to 20 km altitude, i.e. 11% of the signal in a vertical pixel leaks into the neighboring pixels, see Wehr et al. (2023) for a detailed description of the mission and the ATLID instrument.

For the cross polar channel the Mie and Rayleigh signals are not separated. After cross–talk corrections and absolute calibration (and ignoring multiple-scattering contributions), the three ATLID channels can be related to the atmospheric extinction and backscatter signals as:

$$
\begin{aligned}
B_R(z) &= \beta_R(z) \cdot exp\left[-2\int_{z_{lid}}^{z}(\alpha_M(z') + \alpha_R(z'))dr'\right], \\
B_M(z) &= \beta_M(z) \cdot exp\left[-2\int_{z_{lid}}^{z}(\alpha_M(z') + \alpha_R(z'))dr'\right], \\
B_{T,\perp}(z) &= (\beta_{M,\perp}(z) + \beta_{R,\perp}(z)) \cdot exp\left[-2\int_{z_{lid}}^{z}(\alpha_M(z') + \alpha_R(z'))dr'\right],
\end{aligned}
\tag{1}
$$

where $B_R$ is the Rayleigh co-polar attenuated backscatter, $B_M$ is the Mie co-polar attenuated backscatter and $B_T$ is the total cross-polar attenuated backscatter. $z$ is the atmospheric altitude and $r(z)$ the range from the lidar. $\alpha_M$ is the combined aerosol and cloud extinction and $\alpha_R$ the atmospheric Rayleigh extinction. $\beta_M$ is the co-polar Mie backscatter, $\beta_R$ is the co-polar Rayleigh backscatter, $\beta_{M,\perp}$ is the cross-polar Mie backscatter and $\beta_{R,\perp}$ is the cross-polar Rayleigh backscatter. The atmospheric Rayleigh extinction and co-polar Rayleigh backscatter are directly related to each other and both depend on the local molecular density.

When the atmospheric state, i.e. pressure, temperature and therefore molecular density, is known, the only remaining unknown parameter in the $B_R$ [Eq. 1] is the aerosol and cloud extinction profile, which can therefore be independently retrieved from the aerosol and cloud backscatter ($\beta_M(z)$) profile. This ability to perform independent retrievals of the backscatter and extinction profiles is the major improvement with respect to that from an elastic backscatter lidar like CALIPSO (Young et al., 2018) or the retrieval of the optical depth in case of ICESaT-2 (Palm et al., 2022) for which an extinction-to-backscatter ratio has to be assigned.

Before the aerosol and cloud extinction are retrieved it is usual to check where aerosol and clouds can be detected, i.e. a feature mask needs to be created as an input to the extinction retrieval algorithm. In an noiseless and well calibrated HSRL system, the detection of aerosols and clouds may be performed by calculating the backscatter ratio, which is the ratio of the total signal with respect to the Rayleigh:

$$\frac{(B_R + B_M)}{B_R} = \frac{(\beta_R + \beta_M)}{\beta_R} \qquad (2)$$

Since both signals depend similarly on the extinction and distance, the exponential term and range dependence cancels out when calculating the backscatter ratio. When the ratio is greater than 1 there are aerosol or cloud particles present in that pixel, when it is equal to 1, the return consists of molecular backscatter signal only. Equation 2 provides a simple direct method for determining the lidar backscatter profile, however, the utility of a direct application is limited in the case of low SNR situations.

In the case of ATLID, the measured signals are not fully separated since the Etalon is not an ideal filter, see Wehr et al. (2023) for the filter and cross-talk characteristics. To separate the contributions of 'Mie' signals in the Rayleigh channel and vice versa a cross-talk correction is applied within the L1 processor (Eisinger et al., 2022; do Carmo et al., 2021). Taking into account the ATLID design, and additional issues such as e.g. the large distance to the atmospheric targets, limited laser power and receiver area, but also due to the imperfect Rayleigh-Mie separation by the ATLID HSRL spectral elements (cross-talk) (Eisinger et al., 2022), the SNR ratios are much lower than those generally associated with terrestrial lidars.This means that, in general, averaging of the ATLID signals is necessary in order to apply standard HSRL methods to the ATLID L1 data. When aiming to detect aerosol and thin cirrus cloud regions, the signals would have to be smoothed to the point that most of the information content can be lost if the averaging were to be done "blindly" resulting in averaging strong, weaker signals and clear sky areas together.

An example of the expected ATLID daytime signals is presented in Figure 1 for the Halifax-aerosol scene, over the Caribbean, consisting of an ice cloud, a thick aerosol layer (with an Aerosol optical depth (AOD) $\approx$0.28) and a thin aerosol layer between 4 and 6 km height (AOD $\approx 2.2 \cdot 10^{-2}$). The region is part of the Halifax scene with an enhanced marine aerosol optical thickness (Donovan et al., 2023). In the top panel the model truth input extinction field used to simulate the ATLID signals is shown. The three lower panels in Fig. 1 are the co-polar Mie, the total cross-polar and the Rayleigh channel forward modelled signals respectively. The simulations show the results for daylight conditions taking into account all expected noise sources (instrumental effects, e.g. dark noise and external background noise levels).

Note that the absolute attenuated backscatter values in the clear sky are similar in strength to those in the elevated aerosol layer (Fig. 1, panel 2). Simple thresholding to detect significant detections will not work in this case. The main difference

between the two regimes is that the aerosol field shows a more coherent horizontal and vertical field compared to the high spatial variability in the clear sky regions. The effects of attenuation are clearly visible in the Rayleigh channel below the ice clouds. The ice clouds themselves, are seen in both the co-polar Mie channel and the total Cross polar channel. In most cases it will not be possible to mask aerosol layers on an isolated shot-by-shot basis, i.e. as is visible in especially the elevated layer between 4 and 6 km. To enable the detection of these optically thin layers the data needs to be smoothed, however the backscatter signals strength from different target can differ up to a factor of 100. Any smoothing strategy needs to take this into account to not combine information from strong and weak returns resulting in biased, unrepresentative retrievals for extinction and backscatter.

It is expected that the L1 ATLID attenuated backscatters will be reasonably unbiased i.e. the noise will be random and uncorrelated. In addition, at the resolution of ATLID, cloud and aerosol features are not single pixel entities, but will extend in both the vertical and horizontal directions. The combination of these two features (unbiased uncorrelated random noise and extended particulate signals) point to the use of image reconstruction techniques. These techniques can implicitly take into account information from surrounding pixels. Within the ATLID-FeatureMask processor a combination of two techniques have been applied; the hybrid median method to detect strong features and smoothing of the remaining low SNR data to enable the detection of weak features.

## 2.2   FeatureMask definition and Overview

The FeatureMask main output is a feature detection index ranging between 0 (clear sky) and 10 (likely very thick clouds) based on the ATLID signal probabilities. The mask does not distinguish between different particle types but does intend to separate areas of strong returns, weak returns and clear sky regions. The A-PRO processor (Donovan et al., 2022) digests these inputs and defines signal smoothing strategies based on the defined probabilities to enable the retrieval of extinction, lidar-ratio and depolarization ratio. In table 1, the definition of the FeatureMask is provided with a short and long description of each of the possible values. The meaning and explanation of the values should be interpreted loosely. The description is solely based upon what is to be expected based upon the absolute backscatter signals and their signal-to-noise ratios and not based on e.g. retrieved (unattenuated) backscatter values or depolarization ratios. High values of attenuated backscatter, in general, are associated with optically thick ice clouds and liquid layers, whereas most aerosol cases show lower attenuated backscatter values. The detailed lidar target classification is performed within the A-PRO processor Donovan et al. (2022) by the A-TC algorithm, where the attenuated backscatter levels combined with temperature, extinction, depolarization, and lidar-ratio are used to identify different cloud and aerosol types. The description of how each A-FM feature index value is defined is given in the following Sections.

From a usage point of view, data between 5 and 7 can be smoothed or grouped together when present in neighbouring pixels. Additionally one has to ensure, i.e. by target classification estimation, to not combine cloud and aerosol data together. For any value between 8 and 10 care has to be taken before e.g. averaging with neighbouring points. The same holds for all values below 0. The direct surface return (-3) can be substantially larger than a low level aerosol field. Any smoothing of a (sub)surface point in the backscatter retrieval can have a large impact on the resulting retrieved values and subsequent retrievals

| Value | Meaning | Explanation |
|---|---|---|
| 10 | Dense Clouds | Mie signals are very strong indicating cloud returns from liquid or optically thick ice clouds |
| 8-9 | Dense aerosols or clouds | Mie signals are strong and most likely from clouds, although high optically thick aerosols can get to these values |
| 6-7 | Aerosol or thin clouds | Mie signals from optically thin (Cirrus) or attenuated clouds and aerosol regions should reside here. |
| 5 | Low altitude aerosols | Set to bins for which overlying aerosol features are most likely connected to the surface, |
| 1-4 | Likely clear sky | Low Mie signals, indicative of clear air, differences between these values are due to removed features after additional checks |
| 0 | Clear sky | Very low Mie signals expected to come from clear air |
| -1 | Attenuated | Fully attenuated pixels found through the Rayleigh channel signals |
| -2 | No retrievals | Set in the case of a gap in lidar signals or L1 data not trusted (due to calibration, miss-pointing or otherwise) |
| -3 | Surface | pixel found by the surface retrieval, these may include pixels above the surface affected by the surface returns. |

**Table 1.** FeatureMask definition with the first column showing the true output values from the algorithm. The second column provides a hand waving classification and the third an explanation on how this should be interpreted.

like aerosol type. Likewise, adding clear sky pixel values in the smoothing effort will dilute the smoothed backscatter signals
and can result in similar incorrect type classification. For the retrieval of the weaker features of the FeatureMask a number of smoothing techniques and statistical approaches are employed when determining dynamic thresholds.

The ATLID FeatureMask processor follows a number of operations which will be described in more detail in the sections below and are depicted in Fig. 2. The incoming data is first checked for continuity in time (i.e. that there are no gaps in the L1 ATLID data stream). The standard deviation of the noise levels between 20 & 40 km is calculated to be used as a reference value
and the surface pixels are retrieved (Section 2.3). Based on this, a surface mask is defined flagging all (sub-)surface affected pixels (FM=-3). All remaining lidar signals are subsequently converted to signal probabilities (Section 2.4) and the strongest probabilities are defined as FM=10. Using the hybrid-median method the strong features (FM=7-9) and attenuated regions (FM=-1) are retrieved based on the co-polar Mie and Rayleigh channel signals respectively (Section 2.5). The remaining weaker features (FM=5-8) are retrieved using a combination of iterative smoothing and determination of dynamic thresholds
(Section 2.6). Finally, after all retrieved masks have been combined, a consistency check is performed to detect weak features next to strong features that may have been otherwise missed (Section 2.7).

In the case of EarthCARE, an orbit is divided into 8 frames and within A-FM, each frame is subdivided in horizontal along-track blocks of around 1120 km, containing around 4000 profiles, with an overlap margin between the blocks of 100 profiles (≈28 km). These blocks are processed independently in parallel and the results are combined just before the FeatureMask
is written to the product file. This should ensure that, within each block, enough pixels are available to enable statistical approaches and still have a similar enough local atmospheric state. The size of the blocks is configurable and will be evaluated once actual ATLID data is available.

In the case of missing or invalid L1 profiles, a choice has to be made whether the features on both sides of these gaps can be considered continuous or should be treated separately. In the case of aerosol fields, the spatial correlation lengths have
245 been investigated by correlating backscatter profiles between EARLINET groundstations and CALIOP overpasses (Grigas et al., 2015). This showed a 'fairly good' correlation of around 0.86 within a 100 km overpass radius. In the case of A-FM

it is assumed that weak (aerosol) signals cannot be smoothed beyond a conservative 60 km gap. When a gap exceeds this horizontal length, the low SNR data smoothing operation to detect weak features is performed separately on each side of the gap.

## 2.3 Surface detection

The co-polar Mie backscatter signal from the surface can be significantly higher compared with the atmospheric returns just above the surface. With the horizontal and vertical smoothing of the data required to perform any retrievals for a noisy system like ATLID, all pixels which are contaminated by surface returns need to be masked. This is carried out by creating a surface mask. The starting point for this mask comes from the digital elevation model (DEM ; from the ESA EO-CFI ACE-2 database with an accuracy always better than $\pm 16$ m) height provided with the ATLID L1b data files. The ATLID signal has a vertical resolution of approximately 103 m with a vertical range-bin crosstalk of 11% up to 20 km altitude (do Carmo et al., 2021), i.e. 11% of the signal in a vertical pixel leaks into the neighboring pixels. Secondly, the onboard summing of two consecutive lidar profiles is envisaged as standard during the mission. Both effects can cause surface returns to propagate into the range bin above the actual surface and therefore find their way into the smoothed aerosol signals if not correctly detected and masked out. To ensure that no surface signals affect the smoothing of data, a conservative surface influenced height mask is defined on profile by profile basis. For each profile, first the co-polar Mie peak range-gate is located by searching from the lowest pixel up to 2 pixels above the one in which the DEM altitude is located. This 'surface-peak' has to be greater than $3\times$ the average signal noise in the co-polar Mie channel between 20 and 40 km. If the latter condition is not met, the beam is assumed to be attenuated and the surface pixel is set to the DEM pixel. Once the surface-peak ($i_{surf}$) has been assigned, the backscatter signal in the adjacent pixel above is checked by comparing this to both the surface-value itself, the value at $i_{surf}+2$ and the average of the Mie channel signals between $i_{surf}+3$ and $i_{surf}+8$. The surface height is raised one pixel when all of the following conditions apply:

$$\beta(i_{surf}+1) \quad > \quad 0.75 \times \beta(i_{surf})$$
$$\beta(i_{surf}+1) \quad > \quad \overline{\beta(i_{surf}+3:i_{surf}+8)}$$
$$\beta(i_{surf}+1) \quad > \quad 5 \times \beta(i_{surf}+2)$$

The first condition compares the signal with respect to the surface peak, the remaining two aim to evaluate whether the backscatter in the pixel above the surface is indeed higher than expected with respect to the pixels just above, or are part of a vertically extended above surface feature.

Once pixels are defined as being (sub-)surface, they are no longer used in the subsequent feature detection procedures. The main disadvantage of this could be that, potentially, low and shallow features like fog and blowing snow can be missed and will not be reported as features but as surface. Once any of these features is more vertically extended, the second and third condition are aimed to keep the surface at the highest Mie peak around the DEM value. This part of the processor will be extensively

evaluated once ATLID provides real data and updated to provide the best possible low altitude feature detection while at the
same time ensuring that surface backscatter is conservatively identified.

## 2.4 Converting attenuated backscatter signals to signal probabilities

As was depicted in Figure 1, the dynamic range of backscatter signals can span a number of orders of magnitudes for optically
thick clouds while, for targets such as diffuse aerosol fields the signals approach instrument noise levels. In order to linearize
the scale of the signal strength, signal probabilities are used within the algorithm, which takes into account both the signal
strength and its local noise. In this work it is assumed that Gaussian statistics are a reasonable approximation for the detected
ATLID signals. If this is found not to be the case after launch, and the relevant information is available, this step can be updated
to the correct statistical approach. Both the signals and noise levels for each of the three ATLID channels are available in the
L1b products (Eisinger et al., 2022). A number of error estimates, i.e. total, proportionality, systematic and random errors, are
defined in the L1 file, it is assumed that the random errors used within this processor represent the signal standard deviations.
In this case, the probability of detection can be calculated as

$$P_d = 1 - \frac{1}{\sqrt{2\pi}\sigma_s} \int\limits_{S-\sigma_s}^{\infty} e^{-s^2/2\sigma_s^2} ds \tag{3}$$

where $S$ is the signal, $\sigma_s$ the standard deviation of the signal and $P_d$ the detection probability. This integral can be re-written
using the error function ($erfc$) to

$$P_d = 1 - \frac{1}{2} erfc\left(\frac{S-\sigma_s}{\sqrt{2\sigma_s^2}}\right) \tag{4}$$

To enable the detection of very strong single pixel events, i.e. the direct backscatter from small optically thick liquid (e.g.
Cumulus) clouds, the pixels with Mie signal probabilities very close to 1 (high signal with relatively low noise; $P_d > P_{Mie}^{min}$
where $P_{Mie}^{min}$=0.9999 used for the EarthCARE test scenes) are set within the Feature-mask (FM) to a value of 10 (certain target
detection).

## 2.5 Detection of strong features

The most important task of this part of the algorithm is to correctly detects edges with e.g. no smoothing beyond the features
or cutting corners. This will assist in defining the smoothing strategies used in the A-PRO (Donovan et al., 2022) processor and
ensure that liquid cloud signals will not be mixed with neighboring aerosol regions during signal binning/smoothing operations.
Accordingly, this part of the algorithm relies on the application of an edge-preserving technique known as a Hybrid-Median
(HM) filter (Russ, 2007, Chapter 4). The HM filtering procedure preserves lines and corners that are erased or rounded by
conventional median filtering.

Once the surface pixels are known, a hybrid median (HM) filter is applied to the detection probabilities in the entire image
on a pixel-by-pixel basis. The HM filter spans an $n \times m$ box, where n and m are odd integers greater than 5, equivalent to 1400

meter horizontal and 500 meter vertical boxes, and represent ATLID along-track and vertical range-gate pixels respectively. The size of the median filter ($m$) is configurable through a configuration file and two sizes of filters are applied, $m \times m$ and

$m \times 3$ after which the detected features are combined. The along-track oriented $m \times 3$ box shaped filter specifically targets the detection of horizontally distributed 'thin' features e.g. strato-cumulus decks. This is described in more detail later within this section. In the examples shown within this paper, a value of 11 has been adopted for both n and m. This configuration parameter was shown to provide the optimal value for the detection of cloud edges and the filling of internal small feature-gaps using available test scene data. The operational value will be determined once the actual EarthCARE L1b data has been characterized

and calibrated.

     The value of the centre pixel returned by the HM filter is retrieved by calculating the median values of the two diagonals, the horizontal and vertical rows within this box, using the kernels in Figure 3, after which the median value of those four median values is determined. As this latter median is calculated from an even number of values, the third value of the sorted array (not the mean of two values in the centre) is used. Those pixels either flagged as (sub-)surface or non-valid L1b data are not

calculated nor taken into account in the calculation of the median of neighbouring pixels.

     The HM filter is very effective in removing single noise events and filling small gaps within stronger features. As only median values are used, there are no smoothing edge effects. The hybrid median algorithm is run iteratively five times to ensure that the image has converged, i.e. there are no changes in the image between this iteration and the next. This posterizing of the image (Russ, 2007, Chapter 4), where the pixel values are updated each iteration, is a positive side effect of the HM

method and ensures that regions become more uniform and edges between regions more abrupt. This procedure is performed separately for the co-polar Rayleigh and co-polar Mie signals. The resulting Mie image is used for the detection of strong features, i.e. those pixels with a value above a user defined threshold (within this paper a value of 34% is adopted) are set as a strong feature return resulting in FM values of 7, 8 or 9 depending on the absolute hybrid median pixel value. The co-Polar Rayleigh image is used for the detection of attenuated regions, i.e Rayleigh pixels with a hybrid-median-value $< 40\%$ are set

to be fully attenuated [FM=-1].

     In Figure 3, examples of the HM filters used for detecting the strong coherent features are shown. The pixels connected through thick lines are used for the median calculation for the grey center pixel.

     The only coherent structures which will not be detected using the m×m kernel are structures with a vertical or horizontal width of a few pixels. Particularly, the detection of high optically thick water clouds (supercooled layers, stratus or cumulus)

are at risk since they show up in lidar signals as horizontally oriented 'thin' structures of ≈2 pixel thick before the backscatter signal is completely attenuated. To ensure the detection of these important structures, the hybrid median technique is applied a second time using an $m \times 3$ box ensuring that features of only two pixels thick, e.g. water clouds, are detected.

     The results from the two different HM size filters are compared and the additional features in the $m \times 3$ hybrid median results are added to the $m \times m$ mask. The square mask is considered to be the basic masking routine, as it takes into account both

vertical and horizontal coherence and is capable of filling in larger gaps. If only the $m \times 3$ version is used, the strong features in Figure 1 are still detected but with a higher variability and more 'noisy' behaviour within the features. The resulting converged hybrid median results for the aerosol scene signals in Figure 1 are depicted in Figure 4. The top panel shows the co-Polar

Mie signal probability after converting the co-polar Mie signals using Eq. 4, the middle panel shows the resulting signals after 5 iterations of the hybrid-median filter procedure. The bottom panel depicts the Rayleigh signal probabilities. The Rayleigh

signal probabilities can reach very low values at high altitudes as the signals become low, whereas the relative noise levels do not change. Parts of a profile can only be set to fully attenuated for those pixels below a detected Mie feature. In the example shown here this is only the case at the very start of the scene. The procedure is applied to both the co-polar Rayleigh and co-polar Mie data separately. Using the combination of signals in the co-polar Mie and total cross polar channel has been looked at but was discarded as a viable option. There are three main reasons for this. Even though the detection of depolarizing features,

i.e. ice clouds or dust-aerosols, may benefit from this combination, the SNR of the combined signal will become lower for all other pixels. Secondly, the two channels are calibrated separately which can introduce a local bias for regions in between calibration targets, thirdly the cross channel contains both particulate and molecular backscatter signals which will reduce the feature contrast, especially closer to the surface where the integrated molecular attenuation is highest. The resulting Rayleigh mask (not shown) identifies the regions for which the lidar signals are completely extinguished (FM values of -1), while the

co-polar Mie images are used to detect regions which have aerosol or cloud particles.

## 2.6 Detection of weak features

With the high signal to noise features determined, the next step is to search for coherent structures within the remaining part of the co-polar Mie image. For this we start with the original co-polar Mie probability images ($P_{d,Mie}$; Eq. 4) and create the weak feature probabilities ($P_{HM,Mie}$) image. The first step is to remove all the probabilities where strong features, (sub-)surface and fully attenuated pixels were detected and swap these with realistic 'weak feature' values. To accomplish this, the previous

detected strong-feature pixels are filled in using a linear interpolated value of the averaged signals ($\overline{\beta}_{top,bot}$) above and below the already detected feature within the column. The $\overline{\beta}_{top}$ and $\overline{\beta}_{bot}$ values are calculated using a 5x5 box of the remaining $P_{d,Mie}$ pixels. The sub-surface regions are filled, in a similar manner with the lowest pixel set to the background noise value. The reason for this replacement strategy is to ensure that weaker edges of strong features (not detected by the HM procedure)

are not fully smoothed out by the clear sky pixels surrounding them but have relatively stronger signals towards the strong features.

As the most obvious features are already detected and only a noisy image remains, a very simple convolution method is needed to check for the presence of more coherent features within the noisy data. Within the algorithm, the filled in $P_{HM,Mie}$ 2D-array is iteratively convolved using a horizontally oriented 2-D Gaussian normalized Kernel, where the horizontal and

370 vertical standard deviations (in pixel numbers) can be set by the user. In the ATLID examples provided in this paper values of 11 along-track ATLID pixels $\times 1.5$ vertical range gates are used respectively. The iterative smoothing is performed in Fourier space making each convolution a simple matrix multiplication. For four specific configuration specified iteration counts ($N_{i1}$-$N_{i4}$) the inverse Fourier transformation is performed providing images of smoothed probabilities ($P_{i,Mie}$), where $i$ is the iteration number. Each of these smoothed images is checked for the availability of coherent features. In general, for the ATLID

modelled ECSIM scenes, the useful range of iterations runs from 25 up to 170 convolutions, where the inverse FFT was performed for i=35, 70, 140 and 170. The lowest three retrieved convoluted images are used to detect the medium strong

features (FM values of 7) while any images constructed beyond 150 smoothing convolutions is used to determine very low signal to noise aerosol features (FM values of 6). In the commissioning phase the inverse FFT image numbers will be evaluated and updated where necessary.

As the noise is assumed to be Gaussian, the resulting convolved noisy signals are also Gaussian. The Gaussian nature of the smoothed field is exploited in the next step in the algorithm, where a dynamic threshold is retrieved in order to separate clear-sky from aerosol/cloud pixels. From the different convolved $P_{i,Mie}$ images, the one-dimensional detection probability histograms are calculated and examined. The histogram in Figure 5 depicts the number of pixels within a signal probability bin, normalized to the histogram maximum. The histogram maximum is determined by the clear-sky pixels, since most of the

pixels in the atmosphere do not contain enough aerosols to enhance the co-polar Mie backscatter signals. On the left side of the histogram peak, one finds the pixels which originally had very low or negative backscatter values which have not been smoothed enough. On the right side of the peak, one finds the pixels which have enhanced Mie backscatter values, which could in principle be associated with aerosol or low backscatter cloud returns.

        In Figure 5, two example of $P_{i,Mie}$ histograms are shown (blue lines) after 40 and 80 convolutions, respectively, with the

2D Gaussian smoothing Kernel. In order to separate the noise peak from the feature signals, as depicted on the right side, a multi–Gaussian fit (grey line) is performed on the histogram and from this the retrieval of the clear-sky single Gaussian noise peak (red line). Once the multi–Gaussian fit exceeds the noise peak by more than a configurable threshold value (e.g. 8 or 10) the clear sky versus Mie particle probability threshold is defined, which is depicted by the dashed line. Since the histograms and subsequently the multi-Gaussian fit differ from frame to frame, this Mie particle threshold depends on the local conditions.

This ensures that the final threshold used for the particle vs. molecular pixels is flexible and defined by the data, i.e. background noise, and not on a pre-defined fixed value.

        In Figure 5 the signals from the low SNR pixels start with a shoulder on the noise peak (Signal probability ≈0.25) and shows larger deviations beyond ≈0.3. When the shoulder exceeds the noise peak by a factor of 10 (ratio of the grey and red lines) the threshold for this segment of the observed frame is determined. All pixels which make up the area on the right side

of this threshold are defined as part of a feature. Since this is a statistical approach, and holds for individual pixels, it means that neighboring pixels will not per definition be selected. However, in practice, due to the smoothing, neighboring pixels will have similar values and the procedure tends to fill in complete e.g. distributed aerosol regions.

## 2.7 Combining the Strong and Weak Features

As described in the previous sections, the detection of strong and weak features have been addressed using very different

procedures. This may, result in potential gaps between these two types of detected features, e.g. there may be a small gap between a liquid cloud layer (detected using the HM filter approach) and the surrounding aerosols (detected using the Gaussian smoothing filter approach). A second problem which can arise, are small gaps in the detected weak features due to the statistical nature of the approach. A third issue has to do with the wavelength of ATLID. Since the molecular attenuation in the UV becomes significant at low altitudes due to the increasing atmospheric density profile and the relatively large molecular

Rayleigh extinction cross section at 355-nm, the signal to noise ratio of ATLID signals can become especially low close to the

surface even without clouds being present at higher altitudes. In those cases where a layer of weak features is detected close to the surface, it is extended to the surface with a lower FeatureMask index level (FM=5). Finally, there is the chance of a gap occurring between strongly attenuating features like ice and liquid clouds in the Mie signals and the detection of attenuated areas assessed using the Rayleigh signal. In this case, the attenuated region is extended upwards to the lowest pixel with a FeatureMask value $\geq 6$ within each column.

To deal with the issues just described, a combined feature mask (CFM) is created combining the weak and strong feature mask. The merging is performed by applying the hybrid median m×m routine iteratively five times on top of the retrieved combined feature mask. The resulting mask is compared to the original CFM. All features filled in due to the hybrid median filtering are added to CFM, all features disappearing, for FM between 5 and 7, receive a penalty of one to three points bringing them in the range between FM=[1,4] on their detection status.

The final retrieved FeatureMask for the signals, shown in Figure 1, is shown in Figure 6. In the top panel, the FeatureMask output is shown, the bottom panel depicts which part of the algorithm the results originate from. The thick marine aerosol layer as well as the thicker ice clouds at the start of the scene are found mostly using the HM filter step. The thinner elevated aerosol layer is found by the weak feature smoothing part of the algorithm (convolution images 1 to 4). The white contour lines in the top panel depict those regions for which the model-truth extinction values are equal to $1 \times 10^{-6}$ m$^{-1}$. As can be seen that the upper edge of the model extinction field is closely followed in most cases (except below the ice cloud and the upper right side).

## 3 Algorithm performance and sensitivity

Over the years many verification metrics have been devised for comparing forecasts with observations that are commonly used in the field of meteorology. In this section, the verification indices or scores used for the verification of the FeatureMask are described. Details of some of these methods can be found in Fuller (2004). When looking at a forecast event that either occurs or does not occur, the events can be represented by a 2x2 contingency table. Each individual event is categorical, non-probabilistic, and discrete. Examples of this type of forecast include rain versus no rain, or a severe weather warning. In this case it is the detection of a feature (forecast) versus pixels where the input model does have an extinction $> 10^{-6}$ m$^{-1}$ (observed). From the appropriate contingency tables a number of statistical properties can be calculated like the percent of forecasts that are correct (PC=0.91; for the scene discussed above), the hit rate (HR=0.68) or false alarm ratio (FAR=0.02) and more complex combinations of the latter like the Heidke Skill Score (HSS=0.74 Heidke, 1926). The percentage of detected pixels for this scene for each different processor step is as follows; direct detection: 0.1% (Sec. 2.4), hybrid-median: 52.8% (Sec. 2.5), convolution: 34.3% (Sec. 2.6) and combining strong and weak: 12.8% (Sec. 2.7). The fact that significant percentages of detections are supplied by both the hybrid-median and convolution procedures respectively, indicates that both procedures are required.

The detailed ATLID simulations based upon model scenes (Donovan et al., 2023) provide a useful opportunity to evaluate the FeatureMask algorithm in this manner and perform sensitivity tests on the available configuration parameters.

The performance is evaluated in two ways, once looking at the fraction of detected pixels, which have a positive extinction due to cloud and/or aerosols and once the fraction of detected pixels with respect to the actual co-Polar Mie signals. The first relates more to what extinction levels are detectable and the second on how well can we detect features. In both cases the pixels which have been retrieved as fully attenuated have been removed from the sample. In Figure 7, the fraction of detected pixels with respect to the model extinction is plotted in the left panel. A normalized histogram of the model extinction field has been added as reference. The black line shows that above extinction values of of $10^{-5}$ m$^{-1}$, most of the pixels are detected. For higher extinctions the detection rate goes slightly down again. This has to do with the attenuation of higher lying aerosol layers and/or ice clouds which reduces the SNR and therefore detectability of the layers below. The fraction of detection follows the model extinction histogram between $8 \times 10^{-7}$ to $8 \times 10^{-6}$ m$^{-1}$, this is related to the 2D feature detection, it however never surpasses the 50% detection rate in this regime. The yellow dashed line indicates the fraction of missed features above the highest lying detected pixels and depicts the sensitivity of detecting the top of an aerosol layer at particular extinction. Any false detection are shown in a single column at an extinction of $10^{-7}$ m$^{-1}$, since the FAR is only 0.02 and not visible here. In the panel on the right the fraction of detection of all non-attenuated pixels with a positive model extinction are shown. A normalized histogram of clear sky co-polar Mie signals has been added as reference. This histogram depicts the noise of the measurements. Note that features are detected at negative signal levels. This is due to the 2-D smoothing and gap filling techniques applied within the algorithm. The fraction of detection is greater than 50% for signals $>$-$10^{-6}$ sr$^{-1}$m$^{-1}$. Finally a scaled histogram of all signals of the attenuated pixels is provided. This again follows the distribution of the clear sky noise indicative that no usable signals can be found in these. The not detected pixels and false detected pixels follow the clear sky histogram as well, again indicating why these pixels were not detected with respect to the background noise. There may be some room of improvement on the right side of the histogram concerning the non-detected pixels with signal strength $> 2 \cdot 10^{-6}$ sr$^{-1}$m$^{-1}$, but this will have to be evaluated using campaign data and after launch in the commissioning phase.

## 4 Algorithm results

### 4.1 Halifax scene

Specific simulated test scenes have been created from model output data in order to test the full chain of EarthCARE processors (Donovan et al., 2023). One of these scenes is called the Halifax scene. The 6000 km long frame starts over Greenland (night time conditions), crosses Atlantic Canada and ends in the Caribbean (day time conditions). The scene starts with clouds over the Greenland ice sheet followed by high backscatter/extinction clouds down to $50^o$ N. A high-altitude-ice-cloud regime starting over Atlantic Canada down to $35^o$ N is followed by a low-level cumulus cloud regime embedded in a marine aerosol layer below an elevated continental pollution layer at around an altitude of 5 km. The aerosol scene discussed earlier is based on the last part of this scene. The cloud information comes from high resolution cloud-resolving model output (Qu et al., 2022) while the aerosol information is taken from the CAMS model (Peuch et al., 2022).

In Figure 8, the Halifax scene is presented in more detail, starting with the model input extinction field. The forward modelled co-polar Mie and Rayleigh field and finally the retrieved FeatureMask following the methods described above.

Most of the features present in the Halifax scene can be directly seen back in the FeatureMask, where the top of the features nicely follow the extinction field values around $\approx 10^{-6}$ m$^{-1}$ except from the very optically thin aerosol layer between latitudes of 63 and 51$^o$N and altitude between 5 and 6 km. By eye, one can distinguish this tenuous aerosol region, but the smoothing routines can not bring the feature with an average AOD of 0.007 (mean extinction $5.9 \cdot 10^{-6}$ m$^{-1}$ and maximum extinction of $9.0 \cdot 10^{-6}$ m$^{-1}$, is too tenuous to be retrieved as a continuous feature and only a few pixels are found. In general the retrieval nicely follows the edges of the features and there are relatively few false alarms (FAR: 0.01), a relatively high hit rate (HR=0.76) and Heidke skill score (HSS=0.81). One thing that is also visible in the Figure is that no strong differences are expected between day and night time conditions for the ATLID data. The percentage of detected pixels for the Halifax scene for each processor step is as follows; direct detection: 4.2% (Sec. 2.4), hybrid-median: 39.6% (Sec. 2.5), convolution: 44.2% (Sec. 2.6) and combining strong and weak: 12.0% (Sec. 2.7). Overall, the strong features comprise 43.8% of all detected pixels, with similar values observed for the Baja and Hawaii scenes (Donovan et al., 2022).

Another way to show whether all features have been detected is by averaging horizontally all pixels that are classified as (likely) clear-sky (FM values 0 to 4) with no detected feature pixels present above. What one should expect is, that in the case of a clear atmosphere, there are no detected particulate scatterers, i.e. the average Co-Polar Mie channel signals should be 0 at all altitudes. The Total Cross-polar profile however includes both the cross-polarized returns from particulate and molecular particles, where the latter is directly related to the local molecular density. The clear sky signal for the cross-polar channel should thus follow a scaled atmospheric density profile corrected for the Rayleigh transmission profile. In Figure 9, the three clear sky profiles for the three channels are shown. The Mie signals oscillate around 0 above 9 km and shows an enhancement of $1.5 \times 10^{-8} sr^{-1} m^{-1}$ between 5 and 10 km. The Rayleigh channel follows the density profile until the increasing molecular density impacts the signals at 355 nm due to attenuation. The cross-polar channel follows a scaled Rayleigh profile indicating that the signal is directly coming from the linear cross polarization from the molecular backscatter. It does show that no features containing ice particles or dust like aerosols have been missed.

Both the co-polar Mie and total cross-polar channel do show enhancements in the lower kilometer, these signals come from regions which have a low co-polar Mie SNR due to the relatively high molecular attenuation at 355nm near the surface. This leads to a lower probability of feature detection. However when averaged over an entire frame there is a positive signal. Between 5 and 8 km, the averaged co-Mie signal shows a small enhancement which originates mostly from the tenuous aerosol layer between latitudes of 65 and 50 degrees (Fig. 8) which was not detected by the FeatureMask procedure. These averaged profiles are standard outputs from the processor and will to be used for checking the cross calibration performed for the ATLID instrument L1b processor once EarthCARE is in space.

As previously explained, one of the main reasons for the FeatureMask is to guide the implementation of smoothing strategies for, especially, optically thin features. Not only the separation of strong and weak features is of importance, but it is also necessary to ensure that no surface signals are mixed with aerosol signals when calculating aerosol optical properties. In Figure 10, a zoomed view is provided for the area around the detected pixels affected by surface returns in the Halifax scene. Shown are the 16 pixels around the detected lidar surface pixel for each profile, indicated by $S$, the retrieved FeatureMask for these pixels and the statistical properties of the surface and adjacent vertical pixels. The lidar surface pixel can in general be

detected by eye in the figure as the strongest signal within the profiles that occurs either at pixel index [S] or [S-1] in case the pixel above the actual surface is deemed to be influenced by the surface backscatter (see Section 2.3 for the description of surface detection). Statistically the pixel above the surface return exhibits a similar attenuated backscatter histogram to the surface pixel and the pixel below the detected surface between $10^{-9} < \beta_{Mie} < 10^{-4} m^{-1} sr^{-1}$, however the latter two also peak at higher backscatter values due to the surface returns. In those cases where the pixels below the lidar surface have a higher absolute value it is assumed that the detected lidar surface pixel is still dominated by the true surface. The lack of strong signals in the pixel above the lidar surface return indicates that these can be safely used when smoothing signals. Note that on the right side of the FeatureMask, over the Caribbean, the low altitude aerosol pixels are indicated in grey. These pixels have been set to a FeatureMask value of 5 due to the overlying aerosol pixels, not by their absolute signals themselves. The atmospheric attenuation at 355nm often strongly attenuates the signals while it is likely that the aerosol field is extended all the way to the surface. By separating these pixels this way, it provides users of the product a means to decide whether the underlying signals should be used for their specific needs or not.

## 4.2   Using Aeolus data for evaluating the A-FM methods.

In August 2018 the European Space Agency (ESA) launched the Aeolus Earth Explorer Mission (Reitebuch et al., 2019, 2020). Aeolus caries an ultraviolet UV high-spectral resolution lidar, the Atmospheric LAser Doppler INstrument (ALADIN). ALADIN, like ATLID, measures the atmospheric backscatter from air molecules and particles in separate channels, however, the ALADIN instrument is optimized to measure the line-of-sight (los) wind profile observations in the troposphere and lower stratosphere. The los wind component is measured by detecting the direct Doppler shift induced by the atmospheric movements with respect to the satellite. The main detection channels aboard ALADIN are referred to as the Mie and Rayleigh channels (the Rayleigh channel itself is comprised of two spectral filter elements). In the Mie channel, the signal is detected in a spectrally resolved manner and the wavelength shift of the particle backscatter can be detected. The molecular signals are detected using two offset Rayleigh channels each covering one of the wings of the thermally broadened molecular backscatter returns. The ratio of these wings is used to measure the Doppler shift of the detected molecular Rayleigh scattering, i.e. the los wind component.

For ALADIN a relatively low range resolution (minimum 250 m up to 2 km) and low number of vertical bins are available (24) resulting in a maximum altitude up to about 20 km in general. The per-bin vertical resolution of the 24 bins can be controlled and changed dynamically. ALADIN emits circularly polarized light and the cross-polarized return signal is not measured only the co-polarized is detected. Next to the los winds, atmospheric optical properties are provided as secondary product, however, being an HSRL, ALADIN is able to independently retrieve the particle extinction, co-polarized particle backscatter coefficients and therefore the co-polarized lidar ratio.

ALADIN data provides an opportunity for testing the FeatureMask algorithm described in this paper. The procedures described within this paper rely on edge detection and smoothing in both the horizontal and the vertical and were not designed to cope with with a small number of vertical pixels with changing resolution within the profile and also between subsequent

profiles. In order to create signals which can be used by the FeatureMask algorithm, the data is first transformed to a constant
vertical resolution grid starting at the lowest altitude within the orbit up to the maximum altitude of the Mie grid for that orbit.

For those pixels that are distributed over multiple vertical bins, within the newly defined high resolution grid, the low
resolution signals have an added random normal component using the errors reported in the Aeolus L1 product. The main
reason for this step is to ensure that single high backscatter returns with relatively large errors do not spread out over a
large number of pixels but have values related to the local error estimates. The high resolution Rayleigh and Mie signals are
subsequently fed into the FeatureMask procedure as described earlier. The resulting high resolution retrieved FeatureMask is
finally downgraded to the original grid adopting the highest retrieved FeatureMask value when multiple pixels are combined
within a low resolution pixel. The resulting procedure has been added to the current Aeolus operational L2a processor (Flament
et al., 2021) as the Feature_Mask_Index from the AEL-FM processor output. It has been providing operational results since
version 3.15 from mid–2022 together with the first version of the AEL-PRO processor, which is the Aeolus version of the
A-PRO (Donovan et al., 2022) processor. This nicely shows how procedures developed for one ESA explorer mission can
be adapted for other missions. In Figure 11 a comparison of the AEL-FM results is shown for a collocated Aeolus overpass
with CALIPSO. The time difference between the two missions is roughly four hours and obviously specific features will have
changed within this time span. However, in those cases where long lived events are present, the FeatureMask results can still
be evaluated against the high resolution CALIOP retrievals. In this particular case, the two satellites fly over the tip of Somalia
(east Africa) towards Yemen on May 1st 2019. There is a thick dust layer up to an altitude of 5 km surrounded by ice clouds.
Over the Indian Ocean (left side of the image a number of liquid clouds are visible with low level marine aerosols. On the right
side the CALIPSO signals and VFM mask (Vaughan et al., 2009) show liquid cloud layers which are not visible in the Aeolus
data. This can either be due to the difference in overpass time or that the two observation sheets are not fully collocated in
space. Both the dust and ice clouds are nicely captured by both the FeatureMask and the VFM mask and can be seen by eye in
the respective L1 images. A number of these cases are currently being examined in terms of both the detectability as well as
the detailed retrieval of microphysical aerosol and cloud properties as part of the Aeolus L2 evaluation.

## 5   Conclusions

The Earth Clouds Aerosol and Radiation Explorer (EarthCARE) mission is a combined ESA/JAXA mission to be launched in
2024 and has been designed with sensor-synergy playing a key role in order to retrieve cloud, aerosol and radiation products. A
system of 17 geophysical algorithms (L2) have been designed to work in a chain to perform the best possible 3D reconstruction
of the cloud and aerosol atmospheric state.

In this paper, the ATLID feature mask algorithm (A-FM) has been described, the main task of which is to separate regions
with particle returns from molecular backscatter regions only. It is the first processor in the ATLID HSRL chain and the only
one providing its results at the native lidar grid. The output FeatureMask enables the ATLID profile retrieval processor (A-
PRO) to design optimal binning strategies to minimize the number of shots required for reaching high enough SNR and ensure

that no clear sky and strong surface or cloud backscatter returns are mixed with tenuous aerosol or ice cloud layers. A-FM has been based on a number of (statistical) image reconstruction techniques.

One of the first steps performed is the detection of the surface mask, which includes all pixels affected by the surface backscatter. The current implementation has been conservative in the sense that all pixels above the surface which have a high enough elevated backscatter signal with respect to the pixels above are classified as surface return contaminated. This may include near surface feature occurrence within a few 100 meters from the surface, i.e. fog and blowing snow. Once enough ATLID data is available an attempt will be made to improve upon the surface mask and provide an improved low height feature detection. Next to this mask, the integrated surface returns are written out which in the future are intended to be directly used in the retrieval of aerosol optical depth (AOD) from the lidar signal reflected from the sea surface (e.g. He et al. (2016)).

The A-FM algorithm has been evaluated thoroughly using the synthetic test scenes (Donovan et al., 2023, ECSIM) and AL-ADIN L1 data from the Aeolus wind-lidar mission. The test scenes allow for a direct comparison of the resulting FeatureMask to the model truth fields used as input to the simulator. These comparisons indicates that the mask has a percentage correctness >90% and is capable of reliably detecting aerosol regions with extinctions $> 10^{-5} \mathrm{m}^{-1}$.

For the Aeolus mission, the A-FM processor has been reformed into the operational Aeolus FeatureMask (AEL-FM) processor which is part of the official level 2a Aeolus processor. The AEL-FM processor contains most of the core elements of the A-FM processor, and its successful implementation and subsequent evaluation based on more than 1 year of data provides good insight into the processor core and its capabilities.

Finally, the A-FM outputs will provide a direct way to evaluate the ATLID channel calibration in the L1b data. For the L1b verification, the average clear sky signal profiles for the three ATLID channels, the Co-Polar Mie, total Cross polar and Co-Polar Rayleigh channel have been created. These profiles will indicate for each frame whether the calibration of all cross-talk parameters has been well performed.

*Data availability.* The EarthCARE Level-2 demonstration products from simulated scenes, including the L1b data, A-FM, A-PRO and A-LAY products discussed in this paper, are available from https://doi.org/10.5281/zenodo.7117115. The Aeolus L2a products are available at https://earth.esa.int/eogateway/catalog/aeolus-l2a-aerosol-cloud-optical-product

*Acknowledgements.* This work has been funded by ESA grants 22638/09/NL/CT (ATLAS), ESA ITT 1-7879/14/NL/CT (APRIL) and 4000134661/21/NL/AD (CARDINAL). We thank Tobias Wehr, Michael Eisinger and Anthony Illingworth for valuable discussions and their support for this work over many years. The CALIPSO images were taken from https://www-calipso.larc.nasa.gov/products/ lidar/brows_images/production

*Author contributions.* All authors of this paper, namely Gerd-Jan van Zadelhoff, David P. Donovan and Ping Wang contributed fairly with regard to the development of the studies that led to the results presented here. They also contributed equally to the writing/correction of the different parts of the paper for which they are responsible.

*Competing interests.* The authors declare that they have no conflict of interest.

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

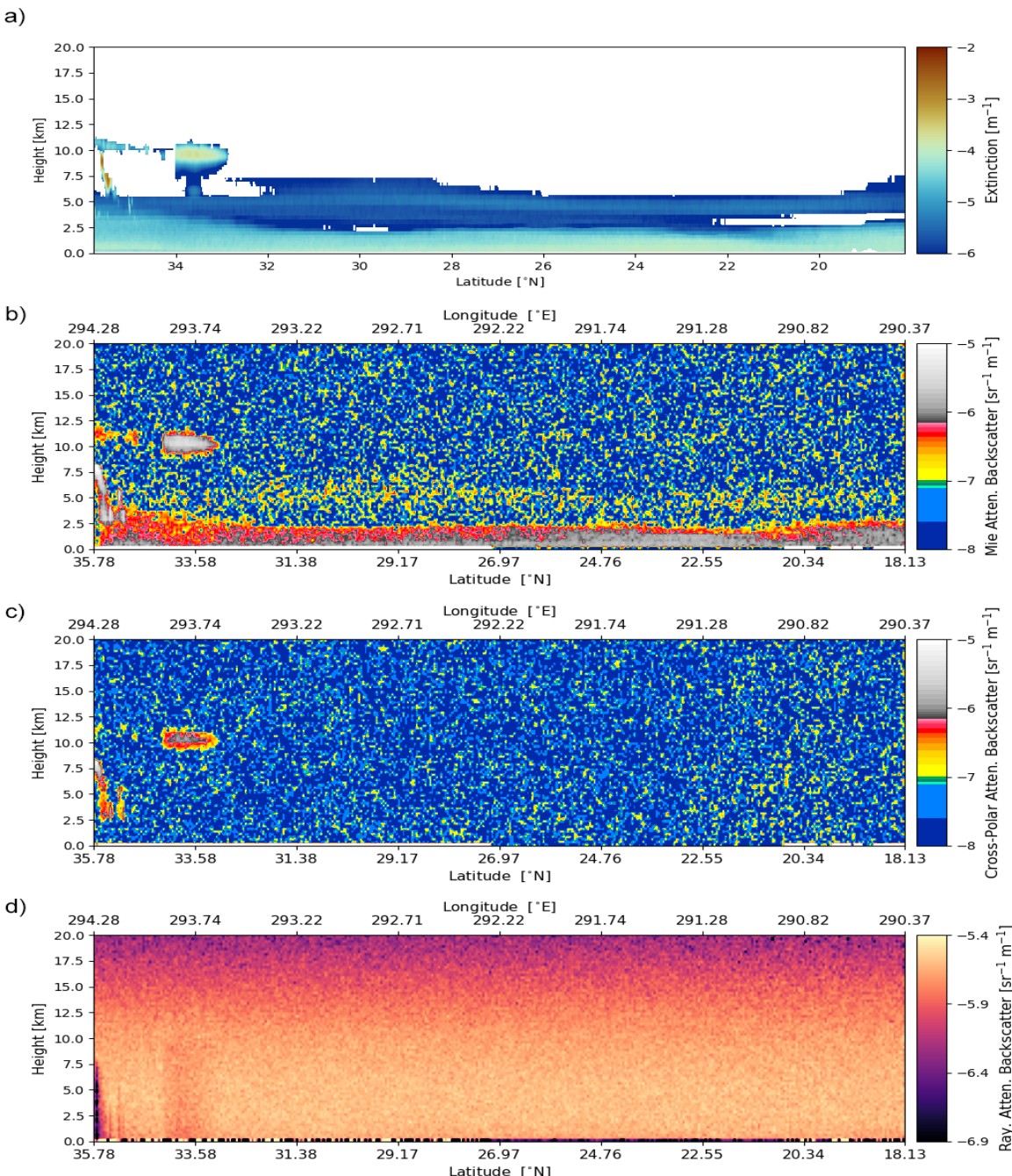

**Figure 1.** The model extinction field used to model the EarthCARE signals (panel a) with the forward modelled ATLID signals for the Co-polar Mie (b), Cross total (c) and Co-polar Rayleigh (d) channels. The scene consists of a thick aerosol layer ($\tau \approx 0.28$) in the bottom 2 km (light green color in top panel), a thin aerosol later between 4 and 6 km ($\tau \approx 2.2 \cdot 10^{-2}$; red to yellow color in top panel) and a few ice clouds at the start of the scene.

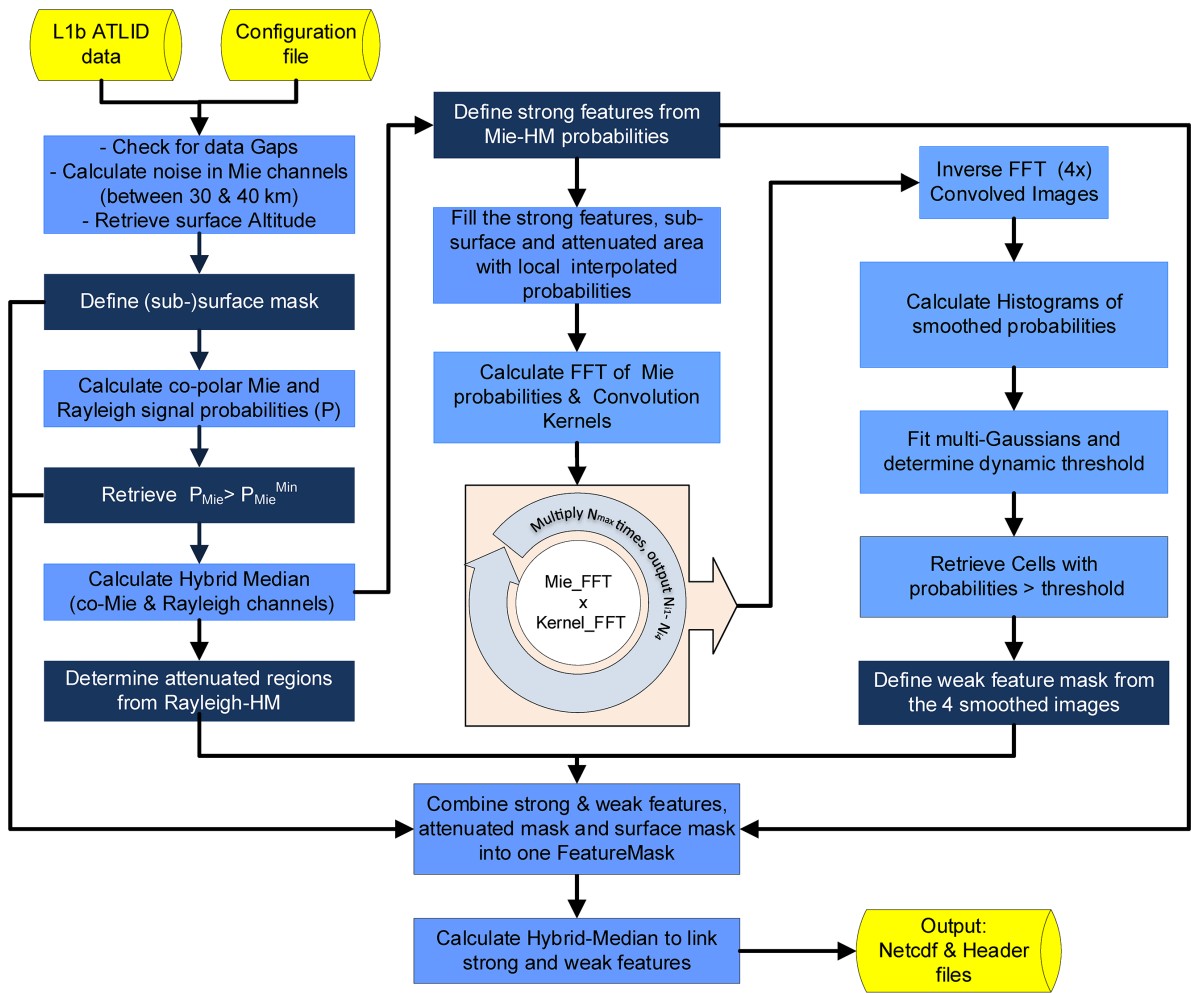

**Figure 2.** Flow diagram illustrating the main A-FM processor steps. Yellow boxes represent input and output files, light-blue boxes the calculations performed and dark-blue boxes the extraction of information from the calculations. The left column depicts the calculation of signal probabilities and the determination of strong features, the center column focuses on the smoothing of the remaining signals and the right columns the subtraction of the weaker features from the smoothed images. Finally all information is combined (bottom center).

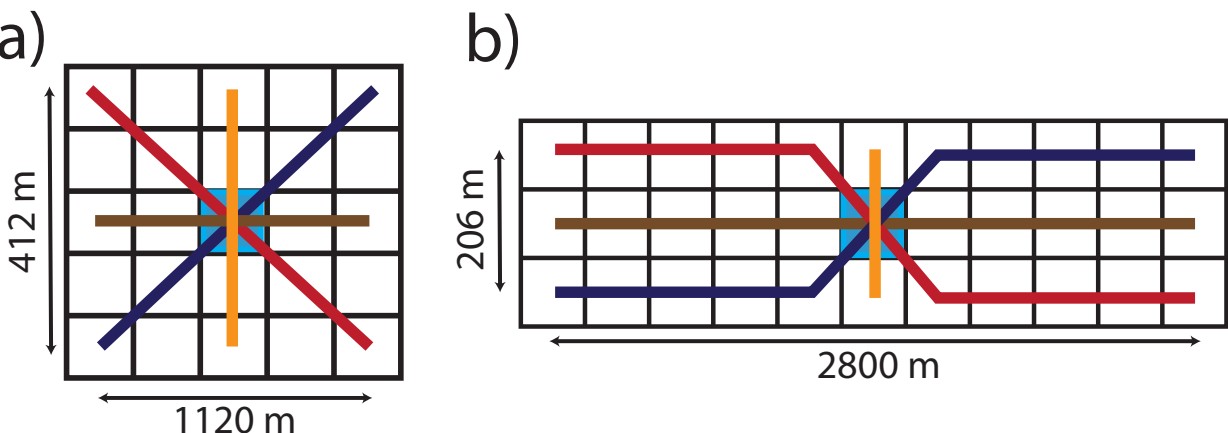

**Figure 3.** Two examples of hybrid median kernels used for finding strong features. The left panel (a) shows a square filter of 5x5 and on the right (b) an example of a horizontal oriented kernel (11x3). The thick coloured lines depict the pixels for which the median values in the center pixel (light-blue) are calculated. The sizes denote differences between cell-centers for the shown filters.

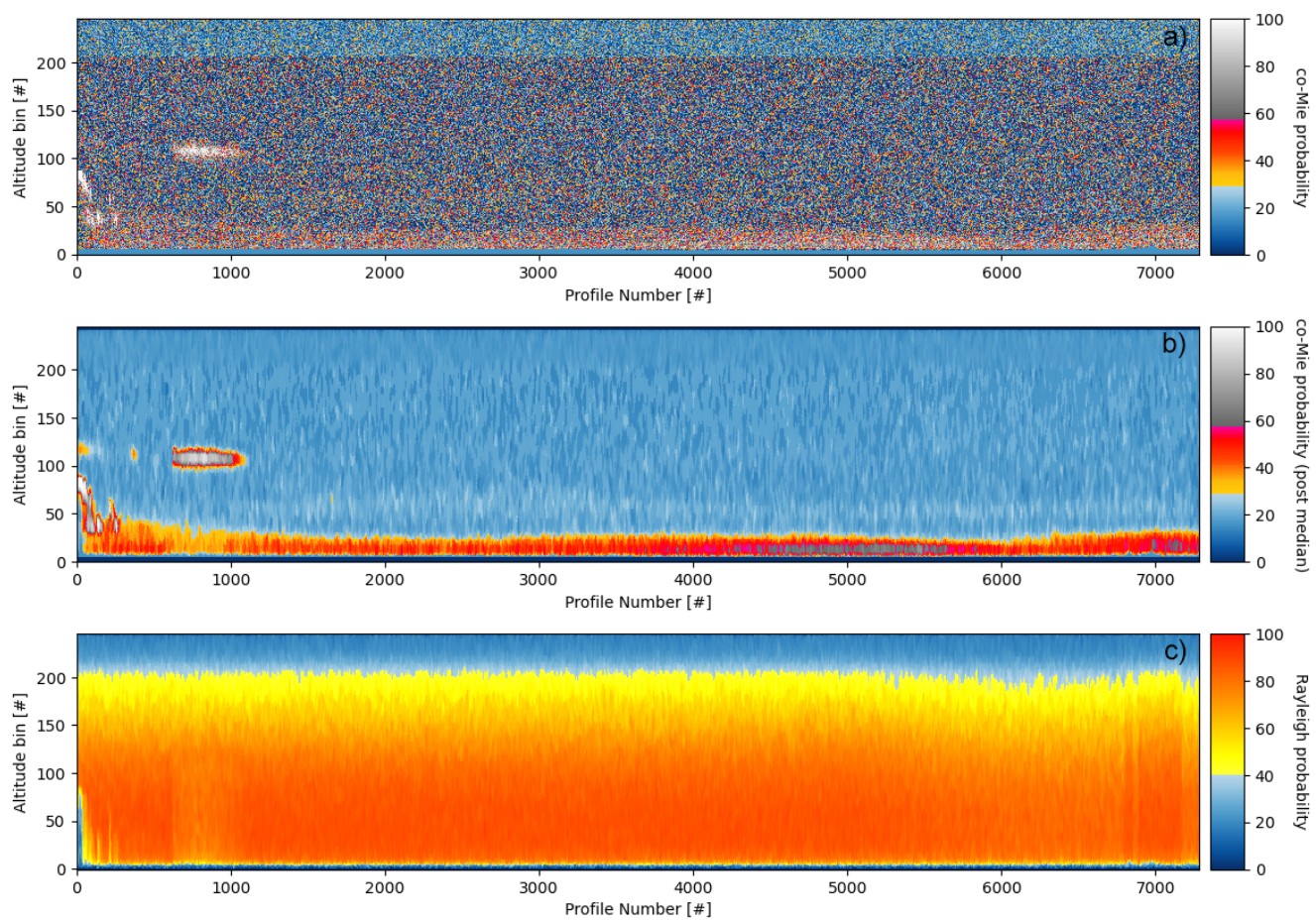

**Figure 4.** The top figure (a) depicts the Co-polar Mie signal probabilities from Equation 4 using the attenuated backscatter signals shown in Figure 1b. The center plot (b) is the resulting image after the HM routine using an 11x11 size square filer, any additional thin-layer features were detected using an 11x3 horizontal oriented HM filter. The bottom plot (c) represents the Rayleigh channel signal probabilities. Note that the figures are shown in pixel number since the procedure is defined in pixel number count and not in SI units of length. The adopted thresholds used for the Mie channel (34%) and Rayleigh channel (40%) are indicated by the change from the blue color pallet to yellow-red.

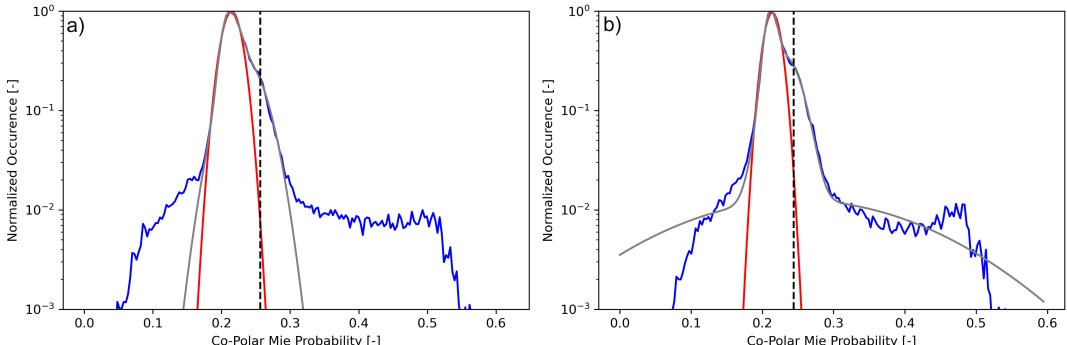

**Figure 5.** Example of two of the probability histograms for one of the regions after 40 (a) and 80 (b) convolutions. The blue lines show the smoothed probability data, the grey lines the multi-Gaussian fit and the dashed black line the retrieved threshold $P_t$. The red line depicts the fit to the central Gaussian noise peak. All pixels with a probability $P_{i,Mie} > P_t$ are retrieved as part of a Feature.

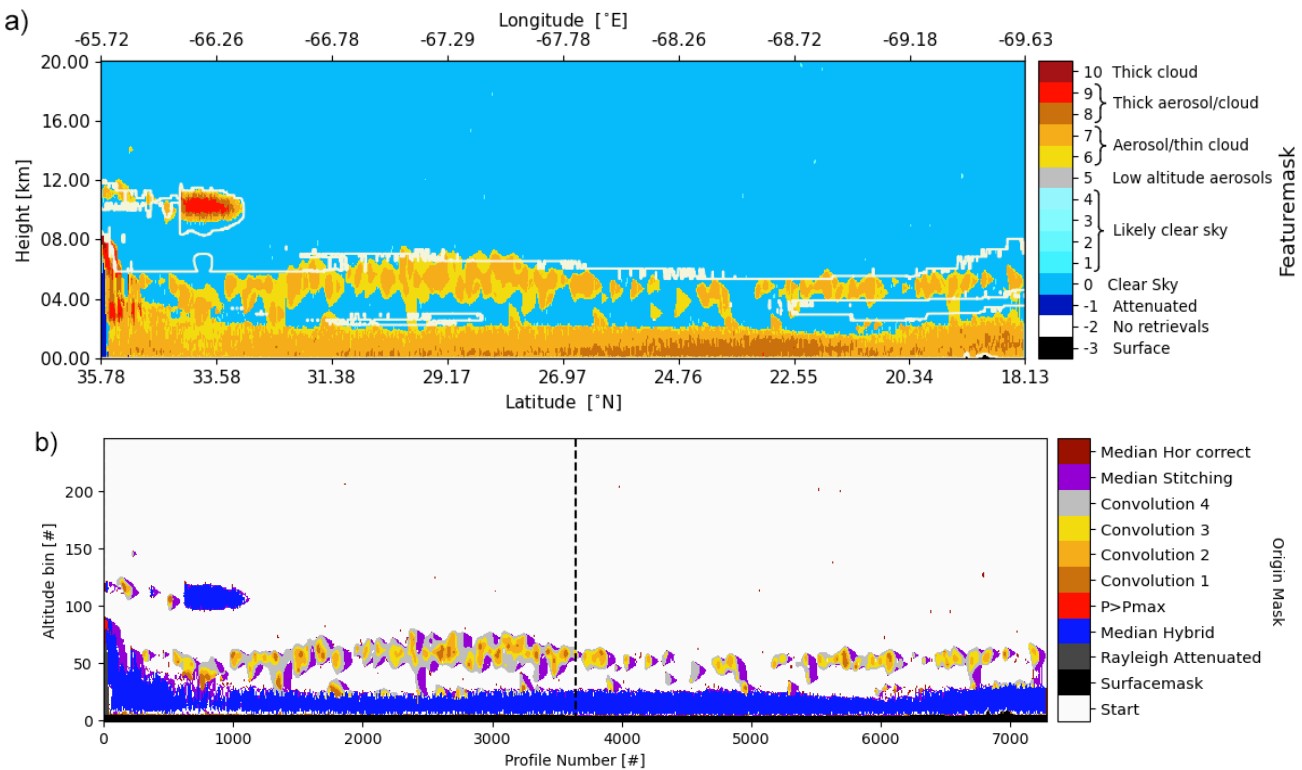

**Figure 6.** FeatureMask results for the attenuated backscatter signals shown in Figure 1. The top panel (a) shows the FeatureMask (filled in contours) with on top the white contour lines depicting an extinction of $1 \cdot 10^{-6}$ m$^{-1}$ of the input model fields. The bottom panel (b) indicates for each pixel from which part of the processor the results originate. Since all procedures are performed in 'pixel' space the lower image is shown in the profile and altitude number count, note that the latter reach the full 40 km height and has not been cut off at 20 km. The dashed line indicates the two regions which were retrieved in parallel by the algorithm.

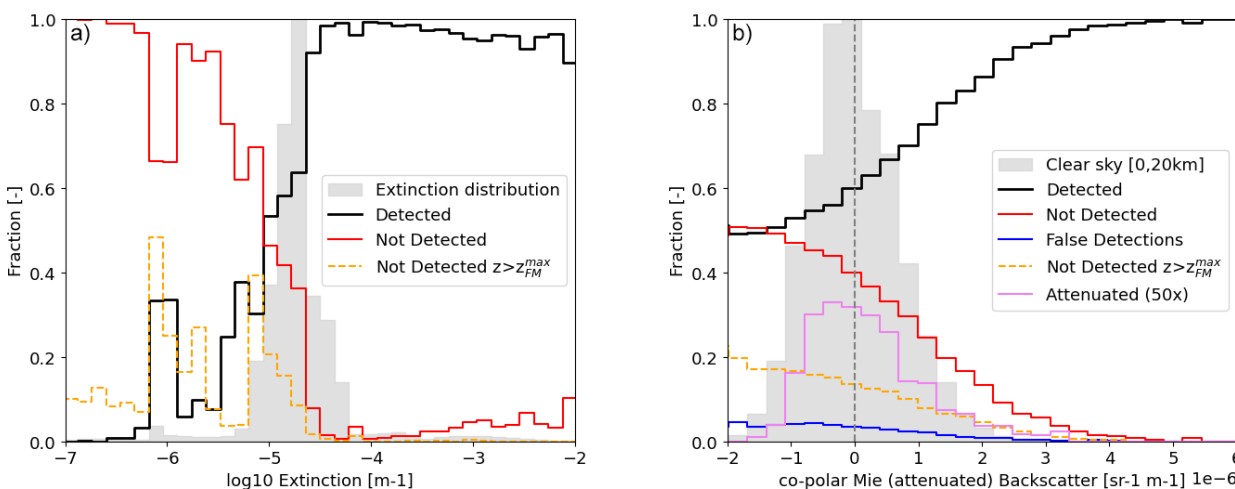

**Figure 7.** Fraction of detected (black-line) and undetected pixels (red-line) with respect to the input model extinction fields which was used to forward model the ATLID L1b signals (panel a). The yellow dashed line shows the fraction off undetected pixels above the highest detected pixel within each column. The normalized model extinction distribution is shown as a grey histogram. In panel (b) the same information is shown with respect to the attenuated Mie backscatter signals with the normalized histogram of the clear sky signals in grey in the background. Additionally the distributions of false detected pixels and fully attenuated pixels are provided.

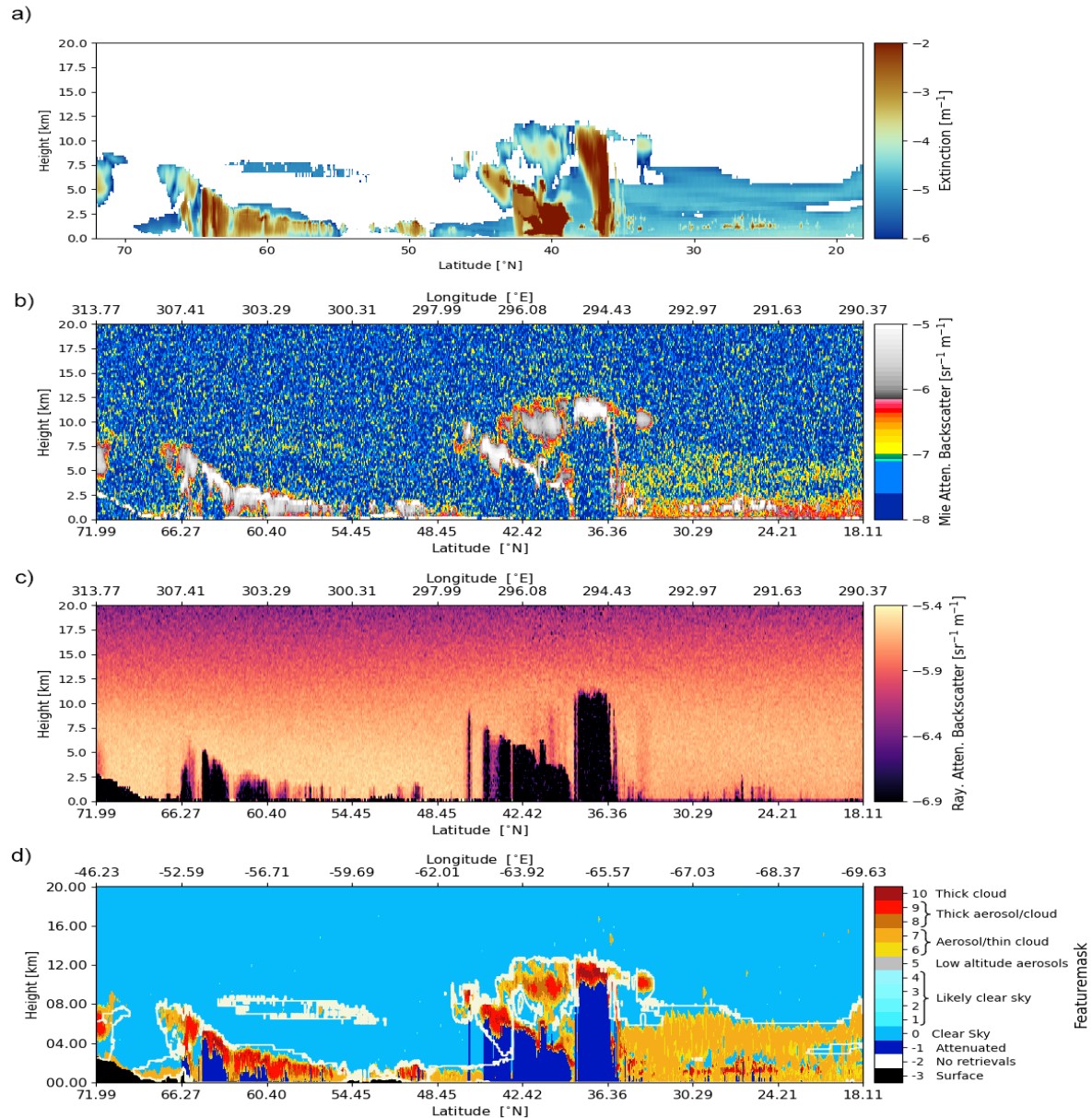

**Figure 8.** Halifax scene; top panel (a) shows the input model extinction field, panel (b) the forward modelled Mie Co-Polar signals and panel (c) the forward modelled Co-Polar Rayleigh attenuated backscatter signals. The bottom panel (d) depicts the retrieved FeatureMask for this scene with on top the $\alpha = 10^{-6}$ m$^{-1}$ model truth extinction contours in beige.

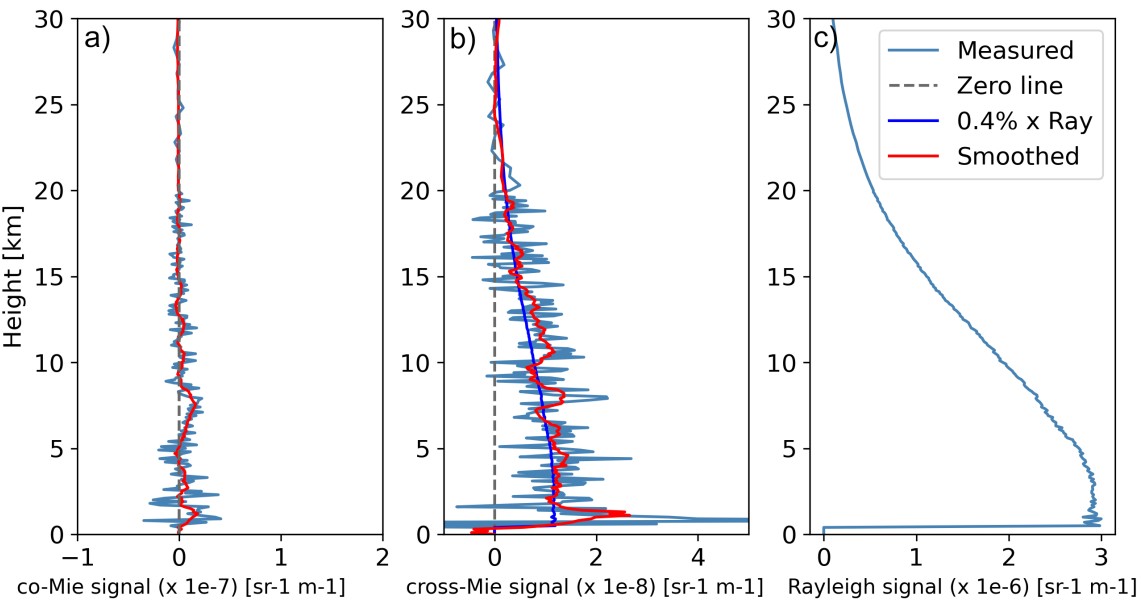

**Figure 9.** Profiles of the average clear sky signals for the Co-Polar Mie (a), total Cross polar channel (b) and Co-Polar Rayleigh channel (c) from left to right. For each plot the average of the entire segment is shown in light blue, a vertically smoothed profile in red and in case of the cross polar channel the scaled Rayleigh profile in dark blue.

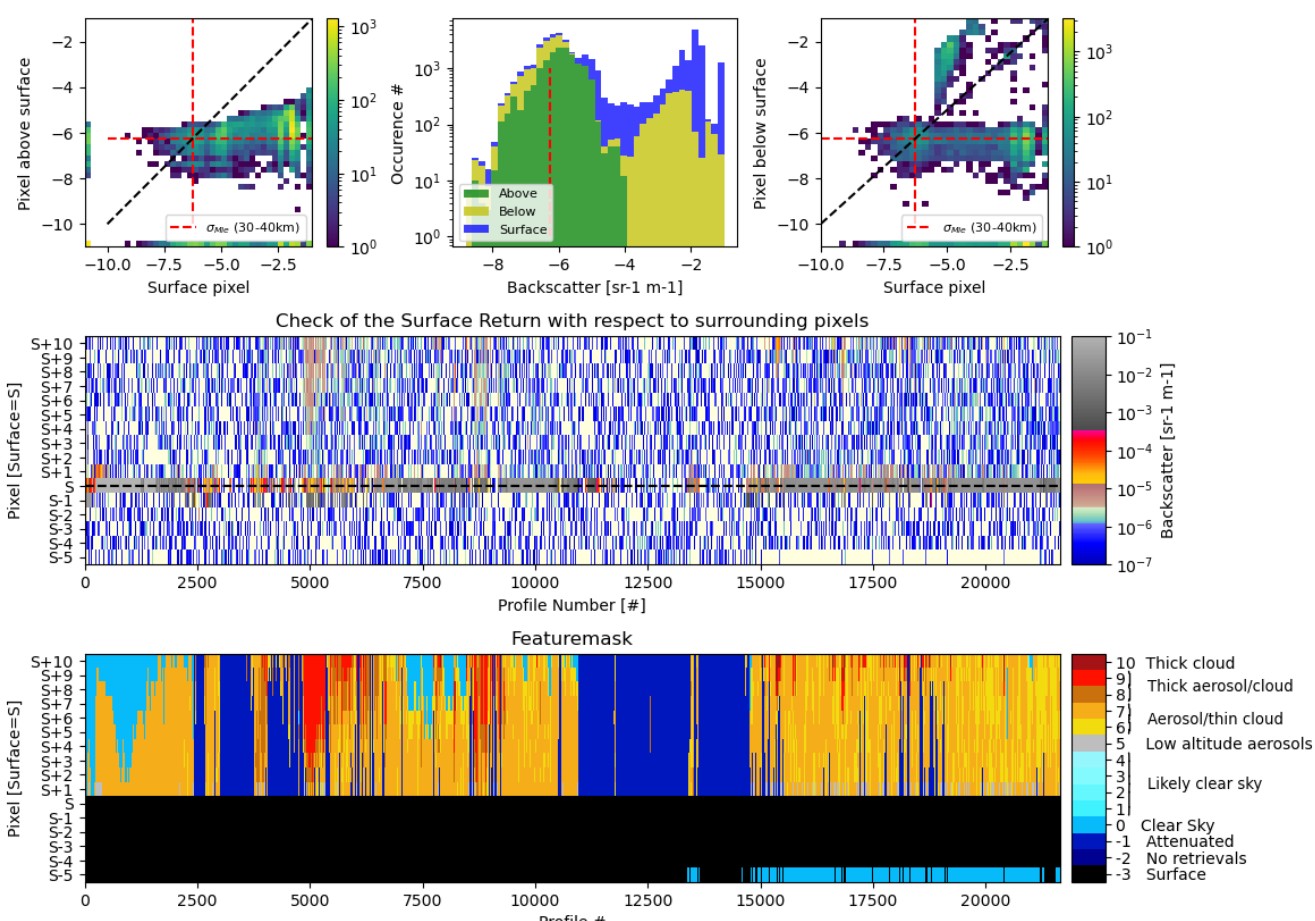

**Figure 10.** Detection of the surface for the Halifax scene following the description in section 2.3. The center panel shows the Mie co-polar returns at native resolution in the 16 pixels around the retrieved surface pixel [S]. The bottom panel shows the subsequent FeatureMask for these pixels. The top three panel provide the 2D histograms of signal occurrence of the surface pixel with respect to the pixel above (left) and pixel below (right), with the dashed red line indicating the co-polar Mie noise levels between 30 and 40 km and the black dashed line the one-to-one line. The top-center panel provides the 1D histograms of signal occurrence for the surface-pixels and the pixels above and below the surface pixel.

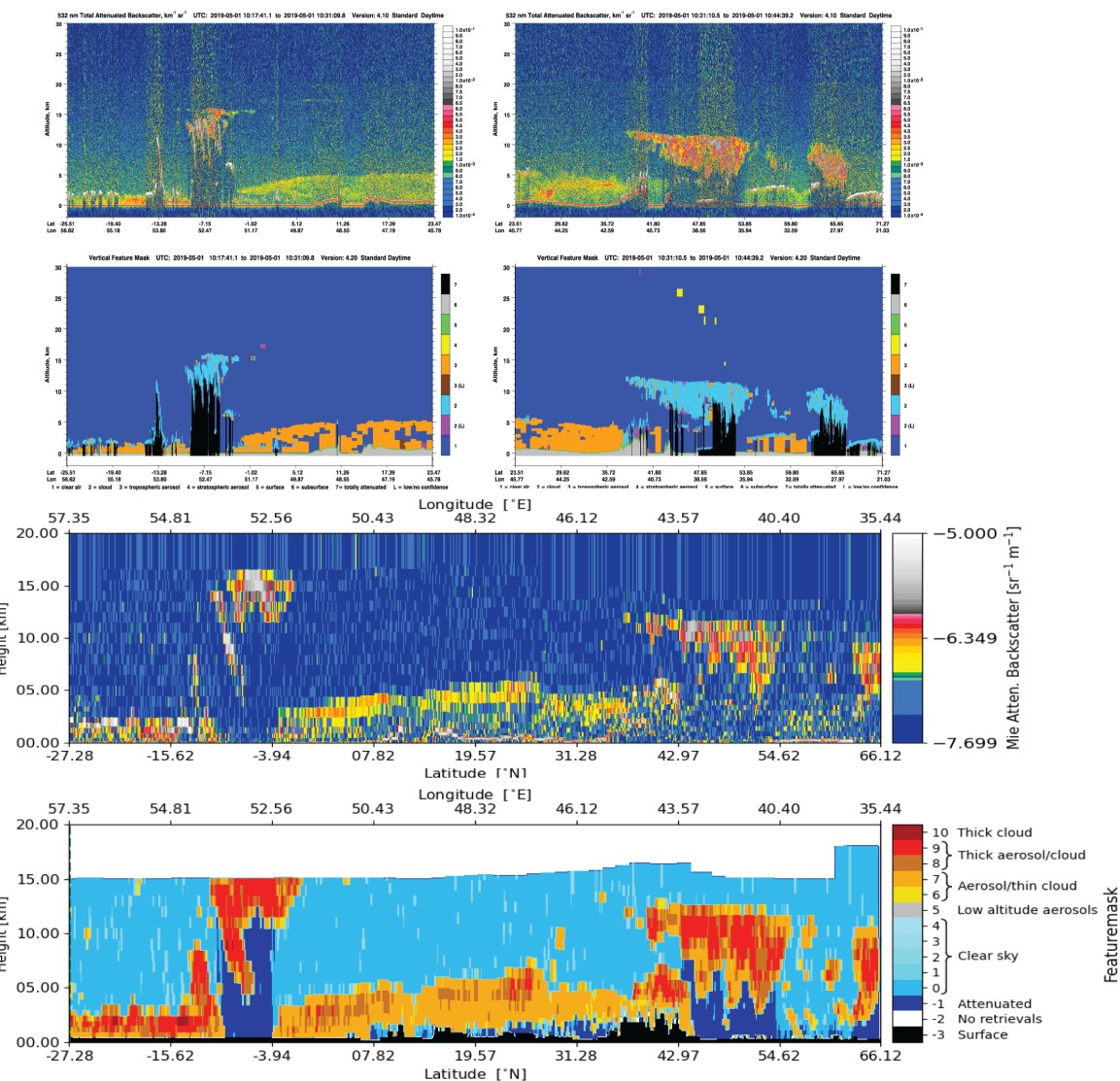

**Figure 11.** Comparison of a CALIPSO and Aeolus overpass (Orbit 3991) on 2019-05-01 over the tip of Somalia (east Africa) towards Yemen. The top panels show the CALIPSO 532nm backscatter quicklooks with the second row the corresponding VFM mask. The third and bottom rows show the 355nm Aeolus backscatter and AEL-FM results for the overpass a few hours later. Both the dust layer and ice clouds are clearly visible in both instrument L1 data and retrieved by their respective feature finders.