# Peer review of "Detection of aerosol and cloud features for the EarthCARE lidar ATLID: the A-FM product"

_EGUsphere, 2023_

## Community Comment (CC1)

**Review of "Detection of aerosol and cloud features for the EarthCARE lidar ATLID: the A-FM product" by Gerd-Jan van Zadelhoff, David P. Donovan, and Ping Wang**

Reviewed by Mark Vaughan (mark.a.vaughan@nasa.gov)

As indicated by the title, this manuscript describes the detection scheme that will be used to identify clouds and aerosols in the backscatter signals acquired by the atmospheric lidar (ATLID) flying aboard the EarthCARE satellite. This is a fascinating approach to a thorny problem, and I commend the authors for developing a novel, sophisticated, and theoretically sound algorithm that appears to work quite well.

In general, I agree with anonymous reviewer #1 on two primary points. First, it is essential that this paper is published, as it will provide the definite reference for one of the fundamental ATLID retrievals and hence will be essential for proper interpretation of all downstream data products. Second, the explanation of the algorithm in the current draft can be difficult to follow, and a top-level diagram (e.g., a flowchart) would be a very helpful addition. Below I have appended an annotated version of the manuscript that contains numerous comments highlighting those areas that I found especially confusing or had specific questions. (I note than some – maybe even many – of these comments come from a practitioner's point of view, and that responses to these questions and comments may not be of interest to more general audiences.)

FWIW, I also have a few other general comments about the manuscript.

o I do not see any attempt in this scheme to correct signal magnitudes for the attenuation effects of overlying layers. I was specifically looking for a discussion of overlying attenuation in section 2.6. Assuming I have not overlooked something blindingly obvious (always a possibility), I'd like to know why this important step is omitted, as, in my experience, failing to do so can lead to missed detections of (e.g.) relatively robust aerosol layers that have been highly attenuated by overlying cirrus.

o I would appreciate more discussion about the magnitude of the crosstalk between the Mie and molecular channels. How effective is the "cross-talk correction applied within the L1 processor"? (Best I can tell, Eisinger et al., 2022 hasn't yet appeared.) Is the residual crosstalk accurately represented in the simulated backscatter signals?

o Expected differences in daytime vs. nighttime performance should be described and quantified.

o Reproducing the authors' findings would, I think, be very challenging based on the limited amount of detail provided in this manuscript. Is there a publicly available Algorithm Theoretical Basis Document that describes the more technical aspects of the algorithm? If so, the authors should cite that document somewhere in their paper. If not, providing this material as supplemental material would also be a viable option.

Here's wishing ATLID and EarthCARE a long and hugely successful on-orbit lifetime!

[revised manuscript text omitted]

---

## Author Comment (AC1)

We thank the reviewer for their constructive comments. We have responded to each comment in-line below, and will soon submit the updated manuscript including the changes described below. With the introduction of a flowchart figure (new Figure 2) all Figure numbers after Figure 1 have been moved up by 1.

The method is difficult to follow. A flowchart would be welcomed to give the reader a snapshotof the algorithm step chain..

We are sorry that this was the case. A flowchart with discussion has been added which will hopefully makes the method easier to follow

More importantly, the equations need to be closely revised as they appear to be incorrect and cause confusion

The equations have been checked and we sadly made errors when converting this from word to Latex. We thank you for pointing this out.

Color schemes used in the figures need a complete revision in order to meet the first requirement for figures of the AMT submission guidelines (https://www.atmospheric- measurement-techniques.net/submission.html#figurestables).

We have revisited the figures indicated in the comments below and improved these.

Below are listed some specific comments. Line 6: "high" > "low"?

Thank you, done

Line 12: What are the "smoothing strategies" for? What is smoothed?

Additional text has been added to provide context.

*"An important goal of the A-FM product is to guide smoothing strategies within down-stream processors e.g. the ATLID profile retrieval (A-PRO) algorithm which directly follows A-FM within the EarthCARE L2 processing chain. Within the A-PRO algorithm, profiles of extinction, backscatter and linear depolarization ratio are retrieved. However, smoothing of the ATLID L1 attenuated backscatter is necessary since the SNR levels present at the ATLID native resolution is generally not sufficient for meaningful retrievals to be conducted. At the same time, to prevent biased retrievals, any smoothing procedure must respect the cloud/aerosol structure and avoid mixing strong features, e.g. clouds, and weak features, e.g. aerosol regions, together. The A-FM product provides the A-PRO algorithm with important information that is used to guide various smoothing procedures."*

Line 16: "ass" > "as"
done

Line 17: "0.9" > "90 %" since you are mentioning "a percentage".
Thank you, done

Lines 59-60: Note that the VFM is a product, not an algorithm. The feature detection algorithm used to generate the VFM is SIBYL (Vaughan et al., 2009). The typing found in the VFM is performed by independent scene classification algorithms: the cloud and aerosol discrimination algorithm (e.g., Liu et al., 2009), the aerosol subtyping algorithm (e.g., Omar et al., 2009), and the cloud phase discrimination algorithm (e.g., Hu et al., 2009).
You are obviously correct and this was misquoted in the article. An update  has been added.

Line 64: Provide a few information on what A-PRO is intended to do. It is unclear what A-FM provides to A-PRO and what A-PRO do with it.
 A description has been added to provide the information

*"Within the A-PRO algorithm the profiles of extinction, backscatter and depolarization are retrieved for which the ATLID attenuated backscatter signal-to-noise ratios are insufficient at the observation resolution. Indiscriminately smoothing signals, within A-PRO, will result in incorrect retrievals which do not represent the actual atmospheric state leading to, e.g. an incorrect target classification. The A-FM product provides the A-PRO algorithm with a field of significant detection. This field is used within A-PRO to decide on local smoothing strategies ensuring that strong and weak signals are not mixed and not diluted by smoothing clear sky values."*

Line 68: "These smoothed signals". Smoothed by A-PRO processor here?Correct, the text has been altered to make clear that this means smoothed data in A-PRO.

Line 75: "NWP" > "numerical weather prediction"
Thank you, added

Line 76: Say what type of signal is the L1b signal.
Added description

Line 79: "FeaturMask" > "FeatureMask"
Done

Line 83: "UV" > "(UV)"
Done

Line 89: "to to" Line 91: "the the"
Done

Lines 95-99: Unclear.
Rewrote the first sentence of section 2.1 to make this more  clear

Line 100: What the difference between this "smoothing strategy" and the "smoothing strategies" in later processors?
In this smoothing strategy the only thing of importance is to find coherent spatial structures. For instance this means that area's where strong features were found will be filled in by local estimates of data based on the probabilities next to the strong features. This adding of 'local averaged data' would be out of the question when performing physical retrievals like extinction. However in this case it enables to fill in weak features next to strong features. The following algorithms use the resulting mask as their starting point of where they can average data depending on additional rules like local depolarization and/or lidar ratio.

Line 120: Eq. (1): Do the Ps represent attenuated backscatter coefficients or power received by the lidar? The r² term should be removed or the lidar constant should be added to the equations for homogeneity.
Thanks for noticing it. You are right the equation is exactly in between the two parameters you mention above. We moved from power to attenuated backscatter in between the ATBD to paper conversion and did this inconsistently. The Ps have been rewritten to Bs to make this more apparent and not related to power.

Lines 146-149: Explain more clearly what does the method consist of (threshold on and averaging of scattering ratio signals).
Lines 146-149 of the original text describes the case where the ATLID signals would

be smoothed without taking into account knowledge of features. This is not done within the ATLID L2 chain at all due to the disadvantages this brings. We rephrased the text to make it more clear.

Fig. 1 top and bottom: Replace the 'rainbow' colorbars by colorblind-safe sequential colorbars (e.g., see https://colorbrewer2.org).
Changed both the top and bottom figures.

Line 151: "AOD" and "tau". Define and use a consistent symbol.
Done

Lines 150-152: Describe the 4 panels. Panel 1 is a theoretical feature mask (the truth we want to retrieved?). Panels 2-4 are the ATLID attenuated backscatter signal channels simulated from the feature mask extinction (and including all the expected noises during nighttime/daytime?).
Updated text like suggested

Line 154: "(1, panel 2)" > "(Fig. 1, panel 2)"?
You are correct; added

Lines 173-174: What are the signal smoothing strategies for? Table 1, Value 0: "cleat" > "clear".
Added a comment regarding the smoothing strategies and updated table

Line 205: Provide DEM uncertainties.
The DEM is provided through the ESA CFI ACE-2 dataset with a spatial resolution of 270m. It has an accuracy of greater than +-16m for the lowest quality pixels well below the spatial resolution of ATLID (~103m vertical)

Lines 211-212: "searching up to 2 pixels above". Surface peak cannot occurs below DEM?
It was intended to read from lowest pixel (well below DEM up to 2 pixels above). So surface can occur below DEM. The text has been updated to reflect this.

Lines 216-219: Stress that ALL conditions need to be verified (if so?).
Done

Line 222: "the pixels above the surface is" > "the pixel above the surface is"
Done

Line 233: "within within"
Done

Lines 237-238: How is the noise level estimated?
These come directly from the L1b input data and are not estimated within A-FM.

Line 240, Eq. (3): I suspect several errors here:
"$^2$" missing on sigma_s below the square root.
The square root did not extend until sigma, however I have ensured that it now written as suggested (extending the square root and introducing "$^2$" for readability.

"S²" > "s²" at the numerator of the exponential exponent.
"sigma_s" > "S – sigma_s" at the integral start.
My apologies, it seems that the conversion from Word (ATBD) to Latex formulae has been messed up on my side in more way than one.

Line 242: "error function" > "complementary error function". Line 245: "very close to 1". Be more specific.
The value used (0.9999) within the EarthCARE runs has been specified in the text. This configurable parameter will be evaluated when real data becomes available in the commissioning phase.

Lines 248-250: Specify the type of signal of the images on which the kernels are applied. Detection probability?
Done

Lines 264-267: Rephrase to make it clear that features are only looked for in Mie and that the procedure is performed to Rayleigh only to define attenuated regions.
Rephrased paragraph to describe the usage of the two signals as suggested

Line 268: "a The"
Reformulated sentence

Lines 271-272: Unclear.
Reformulated sentence

Line 278: "in 1" > "in Fig. 1"? Line 280: "bottom" > "middle"
Fixed

Fig. 3: Use colorblind-safe perceptually uniform colorbars (e.g., see https://colorbrewer2.org). Diverging colorbars from the defined threshold would make sense here, meaning Mie could be centered on 34 % and Rayleigh on 40 %. Those thresholds should be mentioned in the figure caption.
Very nice suggestion, thanks a lot. The color tables have been updated to make these colorblind-save with a clear color change at the thresholds used in the Mie and Rayleigh channels

Why are they low values on the top of Rayleigh? Are those regions considered fully attenuated?
This has to do with the relatively low Rayleigh signals and relatively high noise within the channel. Pixels within a profile can not be attenuated if there is no Mie feature above. A sentence has been added to reflect this point.

Is it possible to have no surface flag and no fully attenuated flag in a same profile?
In principle this can happen. The surface flag is calculated before the attenuation and not updated after this

Line 282: "The combination of signals in the co-polar Mie and total cross polar channel". Do you mean you've tried to apply the procedure on the total channel or on both channels separately and then combining the feature masks? It would be interesting to know why the procedure is not applied on the cross polar channel even with a larger threshold to take into account the molecular contamination (or by adapting the threshold to the expected molecular profile).
The procedure was tested using the forward modelled signals from the EC-scenes and showed lower skill scores when including cross polar signals. Hence the reason for our approach at this moment. We agree fully that it is an interesting idea to keep

the option open for testing this again once real EarthCARE data becomes available. The processor will be updated to make this option configuration parameter dependent so that it is easy testable in the commissioning phase.

Line 287: "The resulting Rayleigh mask (not shown)". Rayleigh mask is simply the blue part (< 40 %) of Fig.3 bottom, correct?
Correct!
"helps identifying" > "identifies" (that the only thing it does, right?)
"those regions". Does it refer to some regions discussed in the previous sentences?
Text has been updated to "identifies the"

Lines 308-313: "four specific user defined iteration steps" and "the useful range of iterations runs from 25 convolutions up to 180 subsequent convolutions". I don't understand.
Updated text which hopefully makes this more clear

Lines 311-313: FM values unclear. Also, were 8-9 detected in the previous section? If so, mention it in Sect. 2.5. Also mention when the -1 value is flagged.
Added to the text and updated the discussion

Fig. 4: Use colorblind-safe colors (not red and green together).
Updated figure

Lines 325-326: "Once the multi Gaussian fit exceeds the noise peak by more than a threshold value (i.e. 8 or 10)". In occurrence?
In probability space

Line 336: "2.6.1" > "2.7"
Updated

Fig. 5: "(filled in contours)". Using a contour function tends to smooth the edges. Since the main goal of this article is to present the feature mask, it would be better to show the true feature mask using a classic pseudocolor plot on the native lidar grid.
Contour lines depicting an extinction of 1e-6 m-1 is shown in white, which is already the color used for "No retrievals". Use another color, e.g. grey, for one of them.
Replace "Median Stitching" green by another color (e.g. light blue).
Updated figure, using beige for the contour line. This is now Figure 6 in the updated paper

Lines 355-356: Why using contour lines of extinction of 1e-6 m-1? Why not using the contours of the truth (feature mask of Fig. 1)?
The contour lines follow the extinction fields of the model truth input files as seen in the top panel of Figure 1. The 1.e-6 was chosen here to reflect the contours of the truth without adding too many overlaying contour lines

Fig. 6: Use colorblind-safe colors (not red and green together).
Updated figure

Fig. 6: Why a limit at 1e-6 m-1 if there are undetected features with extinction < 1e-6 m-1?
The idea behind the 1.e-6 contour line was to provide a quick way to determine the quality of the feature finding.

Lines 381-382: I don't understand. Is this information useful for the study?
The info is indeed of no importance for this work and has been removed.

Fig. 7: Replace the 'rainbow' colorbars by colorblind-safe sequential colorbars (e.g., see https://colorbrewer2.org).
Updated Figure

Lines 408-410: Unclear.
Updated text

Line 414: "ot" > "it".
Done

Fig. 9, middle: Use a colorblind-safe sequential colorbar (e.g., see https://colorbrewer2.org). Fig. 9: Why there are clear sky flags at S-5?
Lines 469-470: "the two observation sheets are not fully collocated in space". Provide more information about the spatial collocation (e.g., a snapshot with the two orbit tracks).
Updated figure
Line 491: "))" –
Done
Line 494: "0.9" > "90 %" since you are mentioning "a percentage".
Done

---

## Author Comment (AC2)

We thank the reviewer for the constructive comments and resulting improvements to the paper. We have responded to each comment in-line below, and will soon submit the updated manuscript including the changes described below. With the introduction of a flowchart figure (new Figure 2) all Figure numbers after Figure 1 have been moved up by 1.

More discussion is needed on how the FeatureMask works in situations where there is significant variation in profile-to-profile attenuation, e.g. when identifying lower-level features below an upper-level cloud that has gaps in it. Are the kernels in the median-filter approach adjusted, or does the method rely on pixel value adjustment only? Is this information propagated to subsequent smoothing algorithms somehow?

The text has been modified based on your, the other referee and the community comments by Mark Vaughan.

The verification section would be better served by calculating the contingency tables based on the signal fields (or attenuated backscatter coefficient) rather than the extinction field, since this provides a more direct comparison of the cumulative effect of atmospheric attenuation on feature detection. Verification before and after combination of the strong and weak feature masks would clearly show the additional benefit of the full FeatureMask algorithm.

This is an excellent idea. We have added an additional panel to the figure to visualize the information with respect to attenuated backscatter signal strength. As well as provided the percentages of detection by the different masking techniques. What is for instance nice to see in this additional panel is that the attenuated fields show similar signals strength as clear sky fields, and that a large number of detected extinction pixels have an attenuated backscatter within this clear sky distribution but are detected based on horizontal and vertical correlation of the signal field.

Lines 25-29: Here, it is stated that measurements from 4 sensors are combined, in order to be compared to measurements from 1 of them. Abstract states that measurements from 3 sensors are combined..

This part indeed read different from what was intended. The line has been rewritten to:

*EarthCARE science is built around the synergistic use of these four advanced sensors (Eisinger et al., 2022), where ATLID, CPR and MSI data are combined in order to estimate the 3D atmospheric properties of clouds, aerosols and precipitation, including their optical and microphysical properties.*

Lines 70: What do you mean by 'effects' here? I assume that you mean that is is not just the mask that is propagated, but also decisions made on smoothing. Using 'decisions made' would be clearer.

Followed the advice and updated text

Lines 95-96: Correlation of what data?

We were discussing the horizontal and vertical correlation of the attenuated backscatter data. Added this to the text.

Lines 115-117: It would be clearer to mention the ideal case here and then discuss the cross-talk, its correction and implications after line 141.

Thanks for the suggestion, I agree that this will make it more clear. The text has been reordered

Lines 348-351: Need to be more specific here. Is this procedure performed for each feature value separately? What is the order of precedence if the process finds overlapping features with different values? Features from which mask receive a penalty, and why do some receive a 2 point penalty?

One side effect of the hybrid median masking is the posterizing of the image (Rush 2007), where the pixel values are updated each iteration. This ensures that regions become more uniform and edges between regions become therefore more abrupt as the edge detection remains strong. By iterating the HM filter we only need to check after 5 iterations whether pixels have been removed or if gaps are filled. When they disappear we want to keep track of potential pixels which need to be evaluated after launch. Since 5 is a reserved number for pixels very close to the ground which are expected to contain aerosols. FM values which has a 6 require a penalty of two in order to skip the 5. However it is also sometimes 3 points (for those pixels where FM was originally 7). The following line has been added.

*All features filled in due to the hybrid median filtering are added to CFM, all features disappearing, for FM between 5 and 7, receive a penalty of one to three points bringing them in the range between FM=[1,4] on their detection status.*

Technical comments

 Line 4: Replace 'state is estimated, which then are used' with 'state is estimated and then used'.

Updated text following your suggestion

Line 18 and elsewhere: Choose one format for displaying extinction and backscatter values throughout the manuscript.

Agreed, these were inconsistent.

Line 30: Replace 'have being developed' with either 'are being developed' or 'have been developed'.

Updated text using ' have been'

Line 35: Replace 'rations' with 'ratios'.

Thanks for noticing this, updated text

Line 38: Explain what Aeolus is here (a satellite wind lidar mission) as this is the first time it is mentioned in the main text.

Added a line referring the Aeolus mission.

Line 66: Replace 'aerosols regimes' with 'aerosol regimes'.

Removed the s

Line 67: Replace 'liquid clouds signals are not mixed with aerosol of' with 'liquid clouds are not mixed with aerosol or'.

Updated text

Line 79: Replace 'FeaturMask' with 'FeatureMask'. Suggest using 'regions' rather than 'areas' - 'areas' imply a 2-dimensional (horizontal) spatial extent, whereas the FeatureMask is of time-height dimensions.

Updated text

Line 89: Replace 'to to' with 'to'.

Removed to

Line 91: Replace 'the the' with 'the'.

Updated text to: *In Section 4 the results for two simulated tests scenes and one Aeolus-CALIOP collocated orbit are presented.*

Line 118: Replace 'correction' with 'corrections'.

Updated text

Line 125: Replace 'depend both' with 'both depend'.

Updated text

Line 172: Replace 'area's' with 'areas'.

Updated text

Line 176: Do you mean 'quantitative' here?

Rephrased the sentence: *The meaning and explanation of the values should be interpreted loosely.*

Lines 206-207: Is the vertical cross-talk pixel-to-pixel only? Or does it extend beyond neighbouring pixels?

Added an extensive description of the lidar signals and cross-talk

Line 240: Should mention here how the standard deviation of the signal is calculated.

Added a description: *A number of error estimates, i.e. total, proportionality, systematic and random errors, are defined in the L1 file, it is assumed that the random errors used within this processor represent} the signal standard deviations.*

Lines 244-246: Do you mean single pixel elements in both the vertical and spatial sense? Not sure that you need to specify 'Cumulus' here as such 'single pixel elements' could also arise from liquid clouds or other features which may be narrow in one dimension (along track) but extensive in the other spatial dimension (perpendicular to the track).

You are correct that it reads to constraining this way. Rephrased the text.

Line 265: For 'Those pixels' do you mean 'Co-polar Mie pixels'?

Both the Co-polar Mie and Co-polar Rayleigh pixels. The line has been rewritten to:

*The resulting Mie image is used for the detection of strong features, i.e. those pixels with a value above a user defined threshold (within this paper a value of 34% is adopted) are set as a strong feature return using FM values of 7, 8 or 9 depending on the absolute hybrid median pixel value. The resulting co-Polar Rayleigh image is used for the detection of attenuated regions, i.e Rayleigh pixels with a hybrid-median-value < 40% are set to be fully attenuated [FM=-1]*

The paragraph has been reformulated

The word Figure has been added

Lines 290-291: Replace 'detects' with 'detect'. What do you mean by 'cutting corners'? It might be safer to write 'artificially rounding the corners of 2-dimensional features'.

This was indeed what was meant. I think I was cutting corners when writing it down. This paragraph has been removed from this part of the text.

Updated text

Figures: Some figures will require their colour scales to be modified in order to meet the journal publication requirements.

The color schemes have been updated

Added f

The line was removed when the top panel was also plotted between 0 and 20 km.

---

## Author Comment (AC3)

We would like to thank Mark Vaughan for his thorough review of the paper and remarks to improve this. From the pdf with comments we have distilled all the remarks relevant for the paper and answered these below.

On the points Mark made at the start of the review:

I do not see any attempt in this scheme to correct signal magnitudes for the attenuation effects of overlying layers. I was specifically looking for a discussion of overlying attenuation in section 2.6. Assuming I have not overlooked something blindingly obvious (always a possibility), I'd like to know why this important step is omitted, as, in my experience, failing to do so can lead to missed detections of (e.g.) relatively robust aerosol layers that have been highly attenuated by overlying cirrus.

We have been thinking about how to deal with this from the moment we started on the ATLID chain. In the end we decided not to do this. The algorithm relies solely on the signal probability. If a signal below a strong feature is attenuated so much that the probability becomes comparable to clear sky (noise) probabilities we could enhance the Mie signals by taking into account attenuation but this should also be reflected in an increase in the local error. The background philosophy here is that if there is no detectable SNR with respect to neighbors it can also not be induced by increasing the signals only. If however there is a region of high snr below a cloud with respect to clear sky it should pop-out in the histogram checking later on. The effectiveness of course has to be checked out more thoroughly in our processor evaluation after launch.

I would appreciate more discussion about the magnitude of the crosstalk between the Mie and molecular channels. How effective is the "cross-talk correction applied within the L1 processor"? (Best I can tell, Eisinger et al., 2022 hasn't yet appeared.) Is the residual crosstalk accurately represented in the simulated backscatter signals?

We have added text to the paper to reflect this topic.

The lidar deployed on the EarthCARE satellite (ATLID) is a high-spectral-resolution (HSRL) depolarization lidar operating at a wavelength of 355 nm. The instrument emits linearly polarized laser pulses at a rate of 51 Hz with a pulse energy of 31-35 mJ. The output beam has a divergence of 36 μrad and points 3o backwards in order to minimize specular reflection by ice cloud particles. The laser beam is collected by a 62 cm telescope and separated into three receiver channels. The incoming signals first pass through a polarized beam splitter separating the cross-polar signals from the co-polar signals. The co-polar contribution in the return signal is subsequently separated into contributions from the thermally broadened molecular (Rayleigh) return and the spectrally narrow elastic backscatter returns from cloud and/or aerosol particles by means of a Fabry-Perot Etalon based spectral filter. The signals from each channel are detected by Memory Charge-Coupled Devices (MCCDs) allowing for single photon detection. The vertical resolution is 103 m up to 20 km altitude and about 500 m up to 40 km altitude, with an effective along-track spatial resolution of about 280 m (after onboard integration of two consecutive lidar profiles). The profile signals will experience a vertical crosstalk of 11% up to 20 km altitude, i.e. 11% of the signal in a vertical pixel leaks into the neighboring pixels, see Wehr et al. (2022) for an elaborate description of the mission and ATLID instrument.

Reproducing the authors' findings would, I think, be very challenging based on the limited amount of detail provided in this manuscript. Is there a publicly available Algorithm Theoretical Basis Document that describes the more technical aspects of the algorithm? If so, the authors should cite that document somewhere in their paper. If not, providing this material as supplemental material would also be a viable option.

There is an ATBD which will be made available in November this year. Around end of September the final version of all L2 processors will be submitted to ESA for evaluation after which they will be implemented in the ECGP ground processor facility. Based on the final changes after testing against Fight Campaign data and further functional testing of the processor, the ATBD will be finalized and be available for the public. We will also take into account the comments received by you and the two other referees to make a more readable ATBD. I hope you are willing to wait for this version of the ATBD until that time.

The following remarks are derived from the comments within the pdf file

it would be helpful if the abstract gave some hint about how the feature detection algorithm works; e.g., is it a moving window threshold scheme, a wavelets-based technique, or something else entirely?

We have made a number of changes in the abstract. I hope these correspond to what you hope for.

and optical?

added optical

are you referring to surface detection here; i.e., "when the surface was detected in the measured backscatter profiles"? saying "impacted" suggests that the surface may have effects elsewhere in the profile, which may be true (e.g., enhanced daytime background) but is still a bit confusing, I think.

It was indeed the second effect I was trying to state. For one, two lidar profiles will be summed on board and therefore the surface may be smeared over vertical pixels

is this supposed to be L1 (i.e., level 1)?

written out Level 1 and defined (L1)

define acronym on first use

added

what is a "frame"?

added description:

To enable the processing of the large data-sets from observation up to L2 retrievals each EarthCARE orbit is separated in to eight frames, divided at latitudes of 22.5N/S and 62.5N/S. There will be a margin on both sides of each frame to ensure that all the profiles within the latitude bands are not effected by edge effects.

what orbit altitude and inclination? what range of latitudes do you expect to sample?

A sentence has been added with the information: EarthCARE will fly in a sun-synchronous orbit, with a descending node crossing time of 14:00 hours, an inclination of 97^o, revisit time of 25 days and at an altitude of 393 km.

do you mean "fully attenuated"? (because the lidar beam is ALWAYS being attenuated, if only by the molecular atmosphere ;^D)

You are correct of course, text has been altered to reflect this

see my previous comment about "impact". maybe I'm not understanding exactly what you mean by this???

rephrased the sentence:

'when the measured backscatter in a range bin that is affected by the ground surface backscatter.'

not quite correct. while there are a number of different *algorithms* for deriving feature masks from the CALIOP backscatter signals, the only data *product* currently distributed by the CALIPSO project is the VFM. because the VFM also includes feature classification info, should also Liu et al., 2019 (https://doi.org/10.5194/ amt-12-703-2019), Avery et al., 2020 (https://doi.org/10.5194/amt-13-4539-2020), and Kim et al., 2018 (https:// doi.org/10.5194/amt-11-6107-2018)

also see Hagihara et al., 2009 (https://doi.org/10.1029/2009JD012344) and Mao et al., 2021 (https:// doi.org/10.1016/j.rse.2021.112687)

The text has been altered following the comments above,

again, not quite correct. the CALIOP layer detection algorithm described in Vaughan et al., 2009 is a "feature finder" that makes no attempt to determine feature type. as with the EarthCARE scheme, layer type determination is done afterward by separate cloud-aerosol discrimination (CAD), cloud phase, and aerosol subtyping algorithms; see, respectively, Liu et al., 2019 (https://doi.org/10.5194/amt-12-703-2019), Avery et al., 2020 (https://doi.org/10.5194/ amt-13-4539-2020), and Kim et al., 2018 (https://doi.org/10.5194/amt-11-6107-2018)

I have to admit that this was indeed slightly different than I had in mind. Thanks for the feedback! Text has been altered following yours and the first referee his/her discussion.

where can readers an image of the entire Halifax scene? if it's provided in this paper (and it really, really should be!), please include a reference to the appropriate figure here.
The reference to Figure 8 containing the entire Halifax scene (used to be Figure 7 in the first version of the paper) has been added

preceding the feature mask discussion with an "instrument overview" section that described relevant measurement characteristics and capabilities would be A Genuine Good Thing[TM]. it would be helpful to know things like laser rep rate, per pulse energy, pointing angle, vertical and horizontal sampling intervals, detector type (e.g., photon counting or analog), etc.

Thanks a lot, very good point. For the EarthCARE special issue it was advised to leave this for the main EarthCARE overview paper. However, I do agree with your remark and have added a short description.

The lidar deployed on the EarthCARE satellite (ATLID) is a high-spectral-resolution (HSRL) depolarization lidar operating at a wavelength of 355 nm. The instrument emits linearly polarized laser pulses at a rate of 51 Hz with a pulse energy of 31-35 mJ. The output beam has a divergence of 36 μrad and points 3o backwards in order to minimize specular reflection by ice cloud particles. The laser beam is collected by a 62 cm telescope and separated into three receiver

channels. The incoming signals first pass through a polarised beam splitter separating the cross-polar signals from the co-polar signals. The co-polar contribution in the return signal is subsequently separated into contributions from the thermally broadened molecular (Rayleigh) return and the spectrally narrow elastic backscatter returns from cloud and/or aerosol particles by means of a Fabry-Perot Etalon based spectralfilter.

Within the EarthCARE terminology, the former signal is referred to as the co-polar Rayleigh return and the latter as the co-polar Mie return. The signals from each channel are detected by Memory Charge-Coupled Devices (MCCDs) allowing for single photon detection. The vertical resolution is 103 m up to 20 km altitude and about 500 m up to 40 km altitude, with an effective along-track spatial resolution of about 280m (after onboard integration of two consecutive lidar pixels). The profile signals will experience a vertical crosstalk of 11% up to 20 km altitude, i.e. 11% of the signal in a vertical pixel leaks into the neighboring pixels., see Wehr et al. (2022) for an elaborate description of the mission and ATLID instrument.

this paragraph omits a whole lot of critically important details (e.g., the methods used to build the "pre-defined convolved images"). I hope that a much more thorough explanation is given later on…
These are all provided later in the text. I have provide references to the sub-sections here so that you know it will be discussed in more detail

which is it: 3 convolved images or 4? and where could readers see these images? perhaps in (yet to be produced) supplementary material that will accompany the published version of this paper?
You are correct that this is not consistent. The used number (3 or 4) can be set by configuration parameter. In all examples 4 have been used. The three is taken from the text. In the A-FM ATBD (to be distributed after November 2023) when the version for the operational processor after launch has been delivered to ESA this is described in more detail

this manuscript should provide a subset of the ATLID specifications in order to give sufficient context for understanding the detection scheme. at a bare minimum, the horizontal and vertical sampling rates should be provided. and if any data averaging is done prior to launching the layer detection algorithm, that too should be described.
This has been added to the description of the instrument including the onboard summing of two consecutive profiles. A-FM will use the resolution of the L1 data provided by ESA and will go to higher resolution if the noise is so low that no onboard summing will be performed.

at some point in the paper, please provide (a) some quantitative measure of the Rayleigh and Mie signal separation; e.g., a contrast ratio or something similar and (b) some discussion about how crosstalk between the channels affects feature detection
a reference to the paper containing all these details has been added

can multiple scattering really be ignored in the layer detection process? IIRC from Reverdy et al., 2015, multiple scattering is a fairly big deal in the ATLID signals.

Obviously this cannot and should not be ignored in the processing of the signals and retrieval of optical properties. These retrievals are performed in the A-PRO processor which does take multiple scattering into account. For A-FM however we are only interested to find areas which show Mie signals (either due to direct backscatter or multiple scattering) and leave it up to A-PRO to separate between the two in the next step.

parallel channel only, yes?
These are indeed only the co-polar (parallel) channels

Young et al., 2018 (https://doi.org/10.5194/amt-11-5701-2018) is a much, much better CALIOP extinction reference.
Followed the advice

AFAIK, backscatter and extinction profiles are not currently being reported in the ICESat-2 data products; see Palm et al., 2021 (https://doi.org/10.1029/2020EA001470)

You are correct, the icesat2 product retrieves optical depth and not extinction. The information originated from the following statement in their ATBD:
The optical depth of the layer is then computed using an assumed extinction to backscatter ratio. (BS_Extinc_Backs) with a nominal value of 25 sr (see Table 4.2). To compute the blowing snow layer optical depth ( τbs ), we sum the calibrated attenuated backscatter within the blowing snow layer and multiply by the product of the bin size and the extinction to backscatter ratio (S).
The CALIPSO and ICESAT-2 lines information were combined in a rewrite and the distinction between the two was lost. Thanks for providing the feedback. The line has been updated

I do not understand the intended meaning here. in a "noiseless and well-calibrated HSRL", why does the ratio have to be "very close to 1" to enable retrievals of extinction and backscatter profiles? certainly the Langley Research Center airborne HSRLs (and, I suspect, other systems too) retrieve extinction and backscatter profiles from aerosol profiles having scattering ratios comfortably larger than "very close to 1".

This was indeed ill formulated and has been rewritten.

I don't understand. how would smoothing ameliorate the crosstalk problem? (and again I wonder: how severe is the crosstalk problem? knowing that would add lots of useful context when discussing topics like this one.)

This is more related to the total noise in the co-polar Mie at native resolution, which includes the crosstalk but also other components. A description has been added with a citation to the paper providing the ATLID details.

does this scene show daytime data or nighttime data?

This is a daytime scene as it was specifically designed for evaluation of the imager (MSI) aerosol retrievals

define acronym on first use

added this in the introduction

not on the full resolution data. but I expect it would work reasonably well with sufficient data averaging, yes?

You are correct that since all liquid clouds have been removed from this test case and with sufficient data averaging a threshold would give a reasonable results in this case.

this is confusing, as there are lots of different noise sources in the signals. do the authors mean random noise here? if instead this discussion is meant to focus on crosstalk, consider replacing the generic term "noise" with the more specific term "crosstalk".

The sentence was indeed not clear. We do use the random noise component within the retrieval and not the total noise. The text has been updated

" Any smoothing strategy needs to take this into account to not combine information from strong and weak returns resulting in biased, unrepresentative retrievals for extinction and backscatter. It is expected that the L1 ATLID attenuated backscatters will be reasonably unbiased i.e. the noise will be random and uncorrelated. In addition, at the resolution of ATLID, cloud and aerosol features are not single pixel entities, but will extend in both the vertical and horizontal directions. "

what is "true noise"? since crosstalk is not really random, but instead a systematic feature of the interferometer/etalon, "bias" might be a better description than "noise"???

see rewrite in the previous remark.

On top of this the Mie channel may experience a height dependent bias when incorrectly calibrated as Rayleigh signal subtraction was too weak or rigorous. The A-FM product is used to evaluate any biases in order to update the cross talk parameters to make the product as bias free as we can

calling this a "probability index" seems confusing; e.g., according to table 1, a "probability index" of 5 indicates low altitude aerosols. is the probability that these are legitimate features actually 0.5? I suspect not. so why not use labels that map into the scattering ratios of the features and indicate how confident users can be that the scattering ratio rises above some altitude-dependent "molecular scattering + noise floor" level? the current set of labels strays into the feature classification regime, which is contrary to statements given in the paragraph beginning at line 57.

It is indeed true that an index based on probabilities is not the same as a probability index. I must have been working on this so long that this became synonyms in my mind. The values between 6 and 10 are closer to this but even then it would not be completely true. I have rewritten the sentence

do you mean "not interpreted in a quantitative sense"?

The line was over complicated and has been rewritten

quibbling here, but..."thick" refers to geometry, whereas "dense" would refer to scattering intensity. one can detect geometrically thick cirrus that has relatively (or even very) low scattering intensity.

You are correct of course. I have updated to dense

I still don't understand what the authors are referring to when they say "pixels above the surface affected by the surface". is this a data averaging concern? please clarify.

This indeed has to do with the onboard summation of two profiles. In the new text in section 2.1 mentions this now

really? I'd think that combining thin clouds and aerosols would be something to be avoided at all costs? for example, consider thin cirrus embedded in lofted Asian dust plumes. these two layer types

have very different lidar ratios (~28 sr vs. ~48 sr), so combining them would yield an extinction solution that would not represent the actual effects of either type.

It was not intended to read as that one can always blindly bin the data. In A-PRO the layer properties (target classification) within profiles is checked before horizontal smoothing is performed. However the retrieval of cloud properties of FM>7 and FM=6 & 7 follows a different path and smoothing within the algorithm.  An additional line stating target classification is needed was added

which signals?

The smoothed backscatter  signals. (added smoothed backscatter)

are these frames approximately equal in size? are data from daytime and nighttime measurements reported in distinctly different frames (i.e., as is done by CALIPSO)?

To enable the processing of the large data-sets from observation up to L2 retrievals each EarthCARE orbit is separated in to eight frames, divided at latitudes of 22.5°N/S and 62.5°N/S. The frames are all 45 degrees in latitude long and not separated in daytime/nighttime.

what is the along-track distance spanned by 100 profiles? what procedures are used to reconcile potentially disparate results within these overlapping regions in consecutive frames?

100 profiles is ~28 km additional on each side of a block of data (~280 m between each profile). All frames overlap with each other on both sides, where the overlap size is defined top- down, from the needs from the last Level 2 processors downwards, to the first processor which in ATLID its case is A-FM, to ensure that each level 2 processors will have enough overlap to perform all smoothing constraints required for that processor. A-FM is the first processor on ATLID side and will have the largest overlap to work with between frames ensuring solid retrievals for each profile within the to be retrieved data.

confusing; is this smoothing part of the feature mask processing or the APRO processing? e.g., see lines 64, 98, and other places throughout the manuscript.

In this case we are discussing the data processing within the A-FM processor to detect low resolution Features. Added "to determine weak features"

which DEM is used? can you provide a reference?

The DEM comes from the ESA EO-CFI ACE-2 database

this is *incredibly important information* that should be given much earlier in the paper.

(if nothing else, knowing about this vertical crosstalk up front would have saved me a whole lot of confusion in trying to understand what was meant by statements like "when the surface has impacted the measured lidar backscatter signals" on line 8)

11% seems enough to blur the edges of dense water clouds.

what's the ATLID pointing angle? is it small enough that you expect to measure the extreme backscatter from horizontally aligned ice crystals? if so, this would be another situation where extra care would be needed in interpreting the signals immediately above.

The angle is 3 degrees backwards. I agree that the information comes late in the paper. The information has also been added to the beginning section 2

what does it mean to be "conservative"? are you erring on the side of false positive identification of surface signals rather than false negatives? what are the consequences of this choice? e.g., for determining the frequency of totally attenuated profiles and perhaps even the altitude of full attenuation (i.e., as in Guzman et al., 2017; https:// doi.org/10.1002/2016JD025946)

We are pushing for as little surface data creeping into the smoothing procedure as possible, this could result in missing the lowest pixel above the surface. We will dive into this issue during the commissioning phase to assess the consequences of our choice and improve this if possible.

the wording here is very confusing.

We have updated the sentence and hopefully this becomes less confusing

this procedure assumes that true surface will never be lower than the DEM altitude. that assumption may fail in polar regions where the surface altitudes are not well known and can be variable (e.g., when ice sheets crack open)

What we actually do is start at the lowest pixel (well below the surface and move up to 2 pixels above the DEM (configurable parameter). Updated the text.

once again the question of orbit inclination and latitudes sample comes up. if ATLID is in a polar orbit where PSCs can be sampled, the "average noise" between 20 km and 40 km could well be misleading.

The inclination angle is 97 degrees. Thanks for the heads up. The polar frames (abs(lat) >62.5 degrees) will be used to define the mean values in this case. I will make sure to evaluate this issue during the commissioning phase.

can you translate this surface detection scheme into an approximate overlying optical depth? that is, what is the approximate optical depth within any column/profile that will cause the profile to be flagged as totally attenuated?

For the scenes tested the optical depth is around ~3 before the column below becomes fully attenuated. There is some spread here as the molecular optical depth is not added in the value of 3. (low clouds will cause full attenuation slightly earlier than high clouds)

a diagram might help here. what are the theoretical (or even empirically derived) justifications for the multiplicative coefficients (0.75 and 5) in these equations? do these conditions perform their intended functions equally well for both strong and weak surface signals?

At this point the values have been designed for the EC-scenes and are all configurable. Even though we think the atmospheric returns and error estimates are close to realistic we do know that the surface treatment is too simple and we cannot verify the absolute backscatters. This will need to be evaluated in the commissioning phase

just to be 100% sure, I went back to double check. and sure enough, there's no assumption of Gaussian statistics anywhere in Vaughan et al., 2009. perhaps the most relevant quotes are from section 2a: "Uncertainties are introduced from a variety of sources, including the stochastic processes governing photoelectron multiplication in the detectors, natural variations in the solar background signals, and

the Poisson-distributed photon arrival rates of the backscattered laser light (Liu et al. 2006)." and with respect to the cartoon in Figure 3 (which is Gaussian) there's this: "The threshold selection problem therefore remains the same, irrespective of the underlying distributions and the amount of averaging applied." on the other hand, section 2 (Detection Theory) of the CALIPSO "Feature Detection and Layer Properties" algorithm theoretical basis document very definitely invokes Gaussian statistics. that would be Vaughan et al., 2005. the full reference is

Vaughan, M. A., D. M. Winker, and K. A. Powell (2005), CALIOP Algorithm theoretical basis document, Part 2: Feature detection and layer properties algorithms, PC-SCI-202.01, NASA Langley Research Center, Hampton, Va 23681, 87 pp. (Available online at http://www.calipso.larc.nasa.gov/resources/project_documentation.php)

At the time we introduced this I was indeed focused on the latter document and did not correctly cite this. I thought it to be most fair to remove the link to CALIPSO in this case, since this is not the assumption being used in the processing.

this is for a single range bin, yes?
correct

check for correctness

Updated the lower limit of the integral and S --> s

how is this determined? is it really the expected standard deviation of the signal in clear skies?

This is determined by the L1 processor designed by industry. Within the L1 processor the errors are separated into total, proportionality, systematic and random errors. The latter is used within this processor. Added this in the text

complementary error function what qualifies as "very close to 1"?

The following has been added (high signal with relatively low noise; Pd >Pmin_Mie. where Pmin_mie=0.9999 used for the EarthCARE test scenes)

what benefits does this hybrid median offer that are not obtained from the simple median used in Hagihara et al., 2009 (https://doi.org/10.1029/2009JD012344)? what I'm hoping to see here is something like a single sentence summary derived from the Rush book; i.e., a very high level overview.

The following has been added:

The most important task of this part of the algorithm is to correctly detect edges with e.g. no smoothing beyond the features or cutting corners. This will assist in defining the smoothing strategies used in the A-PRO (Donovan et al., 2022b) processor and ensure that strong signals from ice /liquid will not be mixed with neighboring aerosol/weak ice regions during signal binning/smoothing operations. Accordingly, this part of the algorithm relies on the application of an edge-preserving technique known as a Hybrid-Median (HM) filter (Rush, 2007, Chapter 4).

how are pixels at the edges of images handled?

Since there is a large margin for each frame these pixels are not retrieved and FM is set to -2 (no retrieval) at the edge.

why 5? it's impossible to translate this number into an along-track distance without knowing the horizontal sampling.

We used 5 to indicate that this part of the algorithm looks at the images as pixels and not per se distance. 5 pixels would relate to 1400 meter (5x280m) horizontal and a bit more than 500 m vertical sizes

the 9 x 3 grid in figure 2 looks to have only 3 segments over which a "sub-median" is calculated; i.e., a red line, a blue line, and a green line.

There are actually two red lines here, one vertical and one horizontal

why? what are the consequences of this choice for the performance of the algorithm? (what I'm hoping the see here is a single sentence summary derived from the Rush book)

When the mean would be taken a new data value is created which could in practice be lowered every iteration, thereby destroying the edge preserving part of the HM filtering. The choice for second or third has a only consequence that the threshold value has to be adapted. The choice for third ensures that the new pixel value is at least as high as the second but in general higher. This is actually a slightly different approach than what is described in Russ Chapter 4 since in that case the horizontal+vertical pixels are ranked, the two diagonals are ranked and together with the centre pixel form an array of three elements. The slightly altered approach used here was used to have a very similar approach for the mxn and mxm HM approaches

do grids require a minimum "valid pixel fraction" in order to be considered in the detection scheme?

Not applied in this case. In the end a threshold is applied to separate the pixels with high enough probability. All others will get their original values before HM to start the weak feature procedure.

IIRC, backscatter coefficient distributions in aerosol layers can be log-normally distributed. since these distributions are skewed in linear space, I'm wondering if the authors have encountered any detection anomalies when applying their method to test data derived from real-world aerosol measurements?

A very good question. We did not yet encounter these at this point, but I admit that we have not pursued this with an enormous amount of effort yet. The positive point for A- FM side is that I need to be as complete as I can and the A-PRO algorithm actually prefers false positives more than failed detections as it will check all the feature flagged data once again before processing. Hence we looked into this but have in our evaluation focused more on this and not with full effort on detection anomalies like you suggest

what parameters are the convergence criteria applied to? are these criteria really so strict that they demand zero changes between one iteration and the next?

No it is definitely not so strict as it read. We do not test if there are no changes after 5 times, but so far I have not experienced that there were visible changes when using more iterations

I'm missing something important here: what is it that changes from one iteration to the next? e.g., are the pixels in each successive HM mask reassigned from their initial values to the median value computed by the median filter? or are the data not initially flagged as features subsequently averaged to some coarser resolution, as in Valliant de Guélis et al. 2021 (https://doi.org/10.5194/amt-14-1593-2021)? of these two possibilities, I suspect the former is the right answer to my question. but I'd prefer to be 100% sure about this (which I would be if the authors included this level of detail in this manuscript)

Indeed the pixel values are reassigned within this module for the 5 iterations. to quote Russ:

If the hybrid median filter is applied repeatedly, it can also produce posterization. Because the details of lines and corners are preserved by the hybrid median, the shapes of regions are not smoothed as they are with the conventional median, although the brightness values across steps are retained. Posterizing an image, or reducing the number of gray levels so that regions become uniform in gray value and edges between regions become abrupt, falls more into the category of enhancement than correcting defects but is mentioned here as a side effect of median filtering.

This therefore helps with using a single threshold as it removes the noise more and more while keeping edges. In our experience no noticable enhancements were seen after 5 iterations

are there any spatial coherence criteria levied on the detection of strong features. that is, are there any algorithmic requirements on the number of vertically or horizontally adjacent pixels that must exceed the 34% threshold?
No, this is the criteria at this point. Smaller features have either been found through the 0.9999 threshold or the mx3 horizontal kernel.

unclear; can surfaces be detected beneath regions where the Rayleigh hybrid median value is less than 0.4?
Yes in some cases (depending of the OD above) the surface can be seen as well as Mie features (i.e. supercooled layers) which have a high enough backscatter in the Mie channel. However the signals of the pixels assigned as fully attenuated should not be used for the detection of thin aerosol/cloud retrievals of extinction etc.

per line 249, shouldn't this be "an n x 3 box"?it really is essential to provide the horizontal resolution of the ATLID data somewhere in this manuscript.
the resolution has been added above (280m). It must be so well known to me that I completely missed this when writing the paper.

I find this somewhat surprising, as atmospheric features typically have a large aspect ratio (i.e., their horizontal extent divided by their vertical depth).
This was maybe a bit over sold. What does happen is that the mxm gives a better contrast between the features as it makes similar regions more uniform. In the end the mx3 gives a nearly as good image but does not always fill in the larger cloud complexes as well. The mx3 does a lot better with the liquid layers ofcourse so it is definitely needed. It is now referred to as the basic masking routine

Is gap filling really a good thing? (I guess the answer depends on the nature and source of the signals in the gaps.)

Depends where you do this. In a mask which tells a retrieval algorithm to have a look at a specific region it is better to have a look at all the pixels within the entire region and not to a spotty version of this. So for the A-FM goals, yes. However only because we know that A-PRO checks all pixels within features once more.

this description is not consistent with what's actually shown in Figure 3, where, according to the caption, "the bottom plot represents the Rayleigh channel signal probabilities".

Correct the text was updated to represent this.

I'd guess this would be especially important for identifying very tenuous layers of lofted Asian dust. are these generally well-detected by the current algorithm?

I am afraid that we have not been able to test this for those layers at this point. But it is a valid point to check for after we have the data to do so.

I'd think that the molecular contribution in the perpendicular (cross) channel would be negligible; e.g., as in CALIOP (see section 4.1.2.1 in the CALIOP level 1 ATBD at https://www-calipso.larc.nasa.gov/resources/pdfs/PC-SCI-201v1.0.pdf). is this not also the case for ATLID?

as you can see in Figure 8 it is not negligible in the UV cross polar channel. Small of course but needs to be taken into account especially with the type classification of low level aerosols.

since it actually is shown, I'll ask these questions: (a) why are the Rayleigh probabilities so very low at altitude bins above 200? and (b) why is the drop-off in the Rayleigh probabilities so very abrupt between bins 150 and 200? (if the Halifax scene is daytime data, I guess the dominance of background noise could be the answer to the second question???)

The probabilities are shown, not perse the mask but I agree the info is there, especially now that we have updated the figures to have a change in colour above and below the used threshold. This is the molecular density profile hidden within the signals. Above 20km (approx. pixel 150) we move to a 500m vertical grid (instead of 100) which exaggerates this. The background noise is indeed stable (of course also here the 500m integration helps but not exponentially).The Halifax aerosol is fully daytime, in the Halifax scene the first part is actually polar night (31 December scene). As you can see we expect way less day- night contrast with respect to CALIOP.

also important not to mix dust and ice, though doing so is perhaps not as disastrous as mixing water clouds and aerosols

I agree with this. We have removed the sentence at this part of the paper (also thanks to remarks of the other referee).

what I don't see is any suggestion that an attenuation correction is applied to the range bins lying beneath "strong" features. why not? it seems that not doing so would degrade the detection of (for example) boundary layer aerosols beneath semi-transparent cirrus with moderately larger (0.5-2) optical depths.

We indeed do not do this. The algorithm relies solely on the signal probability. If a signal below a strong feature is attenuated so much that the probability becomes comparable to clear sky probabilities we could enhance the Mie signals but this should also be reflected in an increase in the local error. The background philosophy here is that if there is no detectable SNR with respect to neighbours it can also not be induced by increasing the signals only.

If however there is a region of higher snr below a cloud with respect to clear sky it should pop-out in the histogram checking later on. The effectiveness has to be checked out more thoroughly in our processor evaluation with real ATLID data.

in terms of backscatter coefficient (or, even better, scattering ratio) what is the approximate dividing line separating "high SNR features" from everything else?

are fully attenuated regions treated the same way as sub-surface regions?

Yes this is done similarly

what is the termination condition that limits the number of convolutions done?

This is pre-defined as configuration parameter. Since we use 4 images including the last one defined. We will need to evaluate the actual parameter values in the commissioning phase Introducing an iterative condition increased the CPU time too much.

does calling these "subsequent convolutions" mean that up to 180 additional convolutions can be carried out after an initial run of 25?

Good point what I meant to say was up to 180. The line is rephrased.

confusing. I do not understand what "the latter two" refers to here.

In the new text I have added explicitly the values for which the inverse FFT is performed. I hope the rewrite makes it more readable.
"For four specific configuration specified iteration counts (Ni1-Ni4) the inverse Fourier transformation is performed providing images of smoothed probabilities (Pi,M ie), where i is the iteration number. Each of these smoothed images is checked for the availability of coherent features. In general, for the ATLID forward modelled ECSIM scenes, the useful range of iterations runs from 25 up to 170 convolutions, where the inverse FFT was performed for i=35, 70, 140 and 170. The lowest three retrieved convoluted images are used to detect the medium strong features (FM values of 7) while any image constructed beyond 150 smoothing convolution is used to determine very low signal to noise aerosol features (FM values of 6). In the commissioning phase the inverse FFT image numbers will be evaluated and updated."

I had lots of trouble understanding the intended meanings of these sentences.

The new text is provided in the previous comment. I hope this makes it more readable.

note to self: since this algorithm works on 1100 km segments of the atmosphere from 20 km to the surface, this assertion should be 100% right on the money

The algorithm works up to the highest L1 data (altitude ~40 km) but for the rest correct

terminology is confusing. as I understand it, the red line indicates genuine signal that is highly likely to originate from molecular scattering. so while this may not be the signal regime you're point to isolate, it's not really noise, is it? also, what is the functional form of the red line? it too is a Gaussian, right???

In the Mie channel the signal in clear sky should oscillate around 0 as the molecular signal should end up in the Rayleigh channel in case of a well calibrated system. Together with the local estimate of the random noise this provides a positive signal probability. I have always dubbed this peak as noise since this should dominate the signal in the Mie channel. The red line is indeed a single Gaussian fit of the Mie clear sky background peak after the multi-Gaussian fit has determined which peak denotes 'noise'. It has been rephrased to clear-sky noise peak

what are the units of this "8 or 10" number? is that 8 or 10 standard deviations of the so-called "noise peak"? (this hypothesis seems inconsistent with the locations of the black dashed lines in figure 4)

No it is the value of the ratio between the observations (multi-gauss fit) and the red line. Units are therefore '-' or '1' depending on your preferences

how is this shoulder identified in the algorithm? and what part of the shoulder is used in determining "when the shoulder exceeds the noise peak by a factor of 10"? looking at figure 4, one might suspect that the algorithm identifies another Gaussian that characterizes the shoulder and it's this peak of this Gaussian that is used for the factor of 10 check.

A fit containing 3 gaussians is fitted first (dark green line). It is forced to contain the first highest 'clear-sky' peak. Based on this a single gauss is fitted containing the peak of the clear sky peak (red line). Now we check the ratio of the data with the red fit and check when the ratio exceeds 10 (dashed black line).

are spatially isolated pixels identified as weak features subsequently removed? or are they retained in the final results?

This is dealt with in the next section. They are retained but will be lowered to FM values between 1 and 5. This ensures that they will not be used but can be traced later on in the evaluation.

CALIPSO does more-or-less exactly the same thing

The idea to do this was definitely inspired by your and your colleagues work

I'd be interested in learning this choice rather than its opposite (i.e., extending the base of the strong feature down to the point where the signal had been identified as totally attenuated).

It basically comes from the fact that the two masks are brought together again at this stage. As you never truly know if you have reached the bottom of the cloud just around the time when the lidar signal runs out of steam or if it has been fully attenuated. We used the detection of any fully attenuated pixels in the same column below as the reason to decide that the latter was the case for those pixels below the feature and the attenuated pixels. For other profiles we let A-PRO decide what to do with these cases.

does this step typically eliminate the "spatially isolated pixels identified as weak features" I asked about earlier in the comment on line 334?

Correct, that is exactly what it does.

are these test results for a daytime scene with daytime noise levels, or a nighttime scene with nighttime noise levels. thinking that I may have overlooked this information earlier in the manuscript, I did a search for the word day. getting no hits on "day", I tried again using "night". again, no hits; i.e., neither word occurs anywhere in the manuscript. this to me is a major omission. the authors simply must address the expected magnitude of algorithm performance differences for daytime and nighttime measurements.

You are correct that this was missing in the text. The Halifax scene is a combination of night and day time data. The Halifax aerosol scene is daytime only. I have added a comment in the discussion of the Halifax scene.

From Greenland to Halifax it is polar night. The MSI-SW channels start to pick up signals just above 55N. Over the Caribbean it is full daylight.

a picky point, but...this is only true if there is no crosstalk between Mie and molecular channels, yes? how well does the simulation capture the crosstalk magnitudes expected in the real signals? in this test, the concern would be about leakage of the Rayleigh signals into the Mie channel, causing the Mie signals to be non-zero (though perhaps so low as to be negligible?)

This is rightfully so a picky point. These profiles have been extensively used in the evaluation of the L1 processor and defined improvements and inconsistencies which need to be corrected in the L1 processor. So we actually use these profiles to evaluate how well the cross talk correction between the Rayleigh and Mie channel has been performed. In A-PRO we have an option to bring this information into the cost function but this will only be adopted if the L1 processor produces very bad quality data.

especially when working at 355 nm, why would you compare against the density profile rather than the profile of attenuated molecular backscatter coefficients??

That is indeed exactly what we do, the reason I used the word density here to link it to something that people can relate or validate to (however I was clearly not complete). The sentence has been changed in to: "The clear sky signal for the cross-polar channel should thus follow a scaled atmospheric density profile corrected for the Rayleigh transmission profile."

the fact that extinction (apparently) will not be retrieved in the range bin immediately above the surface has to be especially disheartening to researchers hoping to drive near-surface PM2.5 from ATLID data.

I agree and until we see how good/bad the surface influence is we think it best to be conservative and retrieve good aerosol retrievals above this bin. Once we have data we will have to do our best to see if we can get more out of this.

while this section provides an interesting look at how well the authors' scheme can be generalized, I'm not sure how germane it is to the performance of the algorithm when applied to ATLID data.

I understand your point. I think this may have to do with the fact that this data was the first real data which could be used. Dave Donovan created new calibrated Mie Rayleigh channels based on the ALADIN Mie channel (which is a spectrum containing Mie and Rayleigh) information. In this procedure the two calibrated output signals have been designed to mimic ATLID data as much as possible so that ATLID processors can be used on Aeolus data.

this explanation of what it means to be "conservative" really should be given earlier in the manuscript (i.e., in section 2.3)
Looking back at your comments I can only agree with you. We have added a comment earlier.

has some kind of averaging or smoothing been applied to the 2nd, 3rd, and 4th panels? I do not see the kind of discrete data points I expect in these plots. note too that the individual panels should be labelled (e.g., a, b, c, and d) for easy reference.

The contour plotting routine may have introduced some smoothing for

if I correctly understand section 2.5, the examples in the paper use either an 11 x 11 grid or an 11 x 3 grid. it's easy to see how the 5 x 5 illustration on the left would scale to 11 x 11. but since I can imagine several other options for an 11 x 3 grid, I think the authors would be doing their readers a kindness by using an 11 x 3 grid in this example.

also,

    -o- consider color coding the lines in the 5 x 5 grid to more readily identify the pixels over which
    "sub-medians" are calculated
    -o- please label figure components (e.g., a and b)

This sounds like an easy and needed thing to do. Will provide updated figures and labelling

---

## Author Response (AR1)

**For the best experience, open this PDF portfolio in Acrobat X or Adobe Reader X, or later.**

**Get Adobe Reader Now!**